# Very-Low-Frequency transmitters bifurcate energetic electron belt in near-earth space

Man Hua [1,2], Wen Li [2✉], Binbin Ni [1,3✉], Qianli Ma [2,4], Alex Green [2], Xiaochen Shen [2],
Seth G. Claudepierre [4,5], Jacob Bortnik [4], Xudong Gu[1], Song Fu[1], Zheng Xiang[1] & Geoffrey D. Reeves [6,7]

Very-Low-Frequency (VLF) transmitters operate worldwide mostly at frequencies of 10–30 kilohertz for submarine communications. While it has been of intense scientific interest and practical importance to understand whether VLF transmitters can affect the natural environment of charged energetic particles, for decades there remained little direct observational evidence that revealed the effects of these VLF transmitters in geospace. Here we report a radially bifurcated electron belt formation at energies of tens of kiloelectron volts (keV) at altitudes of ~0.8–1.5 Earth radii on timescales over 10 days. Using Fokker-Planck diffusion simulations, we provide quantitative evidence that VLF transmitter emissions that leak from the Earth-ionosphere waveguide are primarily responsible for bifurcating the energetic electron belt, which typically exhibits a single-peak radial structure in near-Earth space. Since energetic electrons pose a potential danger to satellite operations, our findings demonstrate the feasibility of mitigation of natural particle radiation environment.

[1] Department of Space Physics, School of Electronic Information, Wuhan University, Wuhan, Hubei, China. [2] Center for Space Physics, Boston University, Boston, MA, USA. [3] CAS Center for Excellence in Comparative Planetology, AnhuiHefei, China. [4] Department of Atmospheric and Oceanic Sciences, University of California, Los Angeles, CA, USA. [5] Space Sciences Department, The Aerospace Corporation, El Segundo, CA, USA. [6] Space Science and Applications Group, Los Alamos National Laboratory, Los Alamos, NM, USA. [7] Space Sciences Division, New Mexico Consortium, Los Alamos, NM, USA. ✉email: wenli77@bu.edu; bbni@whu.edu.cn

Ground-based Very-Low-Frequency (VLF) transmitters radiate emissions at particular frequencies mostly over the range of 10–30 kHz with transmitted power ranging from 20 kW to two megawatts[1–4]. While propagating mostly within the Earth-ionosphere waveguide, which is bounded by the terrestrial surface and the lower ionosphere at altitudes about 90 km, VLF transmitter signals can penetrate through the imperfectly reflecting ionosphere, being guided by the gradients of the Earth's magnetic field, to leak a portion of their power into the Earth's magnetosphere primarily at $L < 3$[5–8] (where $L$ is the geocentric distance in Earth radii of the location where the corresponding magnetic field line crosses the geomagnetic equator). As a result, these transmitter signals, together with naturally occurring plasma waves originating from lightning, plasmaspheric hiss and magnetosonic waves at low $L$-shells, encounter a population of geomagnetically trapped energetic electrons up to ~1 MeV.

Theoretical studies[9–11] have shown that VLF transmitter waves could resonate with energetic electrons and remove them from the Earth's Van Allen radiation belts, a doughnut-shaped region which is known to pose a danger to operating satellites. Early studies provided evidence of the potential transmitter-induced electron precipitation by correlating the electron flux enhancement inside the drift-loss cone with the VLF wave power[12–14]. However, for decades there remained little direct observational evidence that revealed the efficiency of electron scattering by these VLF transmitter waves in geospace[11,15,16], and their contribution to reducing electron fluxes has been prevalently viewed as minor, compared to other natural magnetospheric waves[17–19], since the intensity of VLF transmitter signals is observed to be characteristically weak in geospace[1,3,4]. This discrepancy between observations and theoretical prediction has been caused by a lack of direct relationship between VLF transmitter waves and electron flux variations, owing to insufficient resolutions in previous in situ wave and particle measurements. Using raw flux observations from the Van Allen Probes[20], a recent study showed the bifurcation of energetic electron belt at energies of ~30–130 keV over $L$-shells of 2–3, and attributed it to electron diffusion by VLF transmitter waves through estimates of statistical electron lifetimes[21], but did not provide quantitative simulations for the development of the bifurcated energetic electron belt during specific events. In addition, the frequently concurrent other natural plasma waves[7,22] have made it difficult to quantitatively distinguish the actual role of VLF transmitters in modulating the near-Earth space environment. Understanding the formation of bifurcated electron belts at $L < 3$ is of great importance for understanding the significant role of VLF transmitters in electron loss in the near-Earth space and the feasibility of mitigation of energetic electron fluxes in the natural radiation environment.

Here we report the formation of the bifurcated electron belts at energies of tens of keV, corresponding to the simultaneous occurrence of VLF transmitter waves. Using Fokker-Planck simulations, we show that VLF transmitters effectively remove tens of keV electrons to produce a bifurcated energetic electron belt over $L \sim 1.8$–2.5, characterized by double radial peaks of electron fluxes. Our results provide quantitative direct evidence to link operations of VLF transmitters at ground to changes of the energetic electron environment in geospace.

## Results

**Observations of a bifurcated energetic electron belt**. The spatiotemporal variations of a double-peaked radial profile of energetic electron fluxes were observed at energies of tens of keV at $L < 3.0$ during a 15-day period from 20 February to 6 March in 2016 (indicated by the arrows in Fig. 1c–e), using high-resolution electron flux measurements from Radiation Belt Storm Probes Ion

Composition Experiment (RBSPICE)[23] onboard both Van Allen Probes. The corresponding solar wind conditions and geomagnetic storm activity were at a low level, except that some moderate substorms occurred in between, as reflected in the AE index (Fig. 1a, b). Characterized by local flux minima at $L \sim 2.0$–2.2 and resulting from the decay of electron fluxes at energies of tens of keV, the bifurcated electron belt was distinct from the typical structure of energetic electrons peaking at $L \sim 2.0$–2.5[24] before 12 UT on 21 February. The energetic electron spectra at $L < 3.0$ during three intervals corresponding to three outbound trajectories of Van Allen Probe A (Fig. 1f–h) indicate that the bifurcation of electron fluxes at tens of keV became more evident from 21 February to 05 March in 2016, and the characteristic energy of the major electron flux decay decreased with increasing $L$-shell. It is noteworthy that another instrument onboard the Van Allen Probes, Magnetic Electron Ion Spectrometer (MagEIS)[25], detected the bifurcation of an energetic electron belt similar to RBSPICE but with fewer energy channels (Supplementary Fig. 1). As indicated by the white dashed lines in Fig. 1f–h, the energies of the radial electron flux minima are quite consistent with the minimum first-order cyclotron resonant energies of electrons (see more details in the section of "Calculation of electron cyclotron resonant energy" in "Methods") that interact with in situ observed VLF waves at 24 kHz (Fig. 2a), implying a potentially important role of ground-based VLF transmitters in bifurcating the energetic electron belt.

**Observations of VLF transmitter waves in space**. During the entire 15-day period, VLF transmitter emissions were observed within 10–30 kHz at $L < 3.0$ by the high-frequency receiver (HFR) of the Electric and Magnetic Field Instrument Suite and Integrated Science (EMFISIS)[26] onboard both Van Allen Probes A and B. A representative example on 23 February 2016 is shown in Fig. 2a, exhibiting strong wave power over 18–26 kHz at $L < 2.8$. Two groups of VLF transmitter waves were evident at an intensity level below 5 pT over two different $L$-shell regions (Fig. 2b). One exists at $L < 1.7$, mainly originating from the 19.8 kHz NWC transmitter and 21.4 kHz NPM transmitter, and the other over $L$-shells of ~1.7–2.8 mainly from the 23.4 kHz DHO38 transmitter, 24 kHz NAA transmitter, and 24.8 kHz NLK transmitter[3,4]. Figure 2c displays the radial profile of the root-mean-square (RMS) magnetic wave amplitudes (see the detailed information in the section of "Derivation of various plasma wave amplitudes" in "Methods") of in situ observed VLF transmitter waves averaged over magnetic local time, exhibiting a peak of 4.1 pT (at $L = 2.2$), which exceeds the statistical peak values of VLF transmitter wave intensity during both geomagnetically quiet and moderate periods. This is possibly related to the seasonal effect, since the wave power at $L > 1.7$ mainly comes from the DHO38 transmitter located in Germany, and NAA and NLK transmitters located in North America[3,4] where the transionospheric wave attenuation decreases due to a lower sunlit electron density in February[5]. The comparison of the event-specific wave amplitude with the statistical results during different seasons is shown in Supplementary Fig. 2. Note that the observed VLF transmitter waves are frequently coherent, an example of which is shown in Supplementary Fig. 3. For lightning-generated whistlers[27], plasmaspheric hiss[28] and magnetosonic waves[29], their statistical wave amplitudes under modestly disturbed geomagnetic conditions either decrease or increase monotonically with increasing $L$-shell (Fig. 2d), and exceed or become comparable to the VLF transmitter wave amplitude. However, the energy of electrons exhibiting the most evident bifurcation feature (tens of keV) is close to the first-order cyclotron resonance energy corresponding to the waves at high frequencies (>10 kHz), suggesting the potentially dominant role of VLF transmitter waves in bifurcating the energetic electron

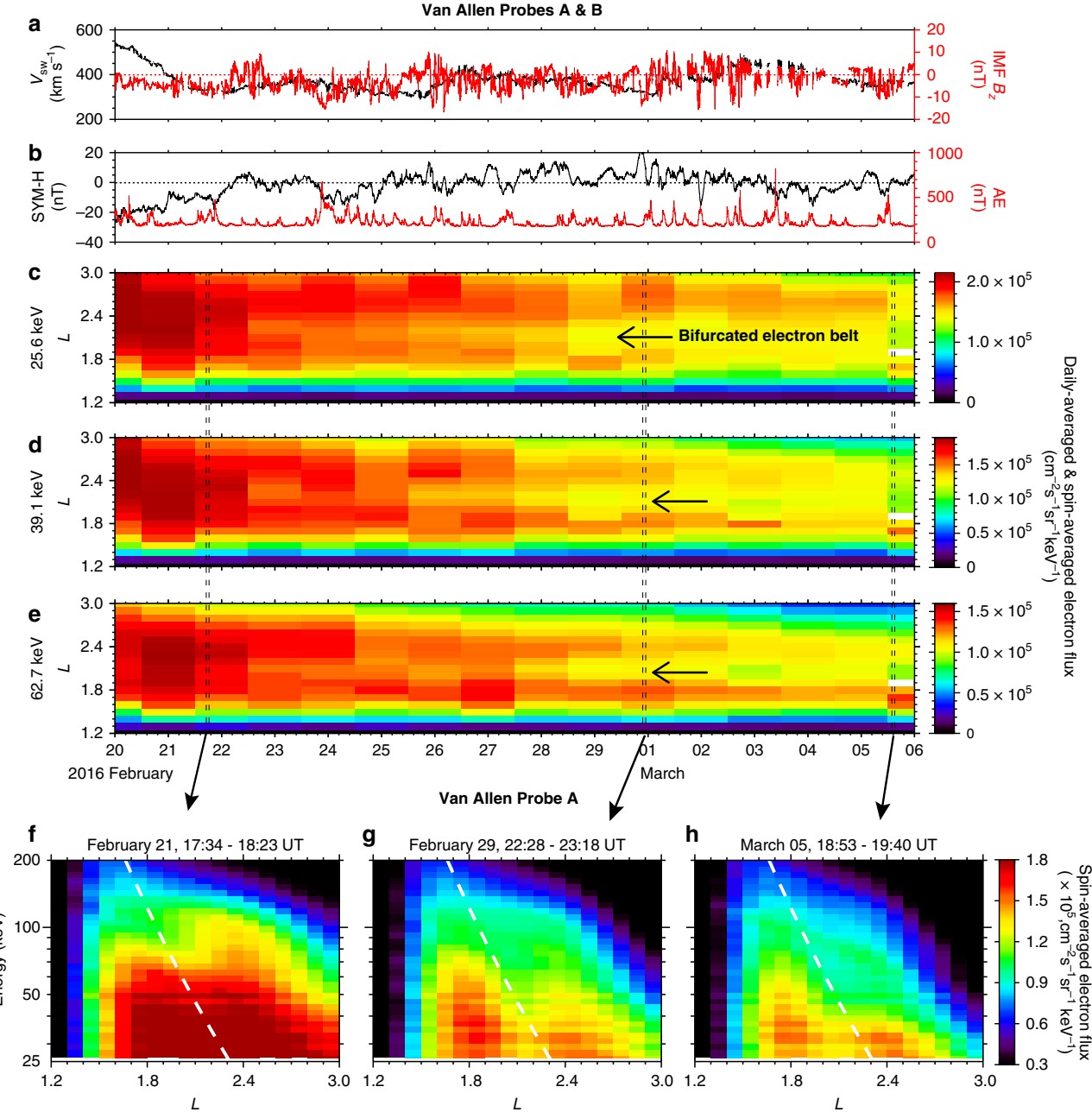

**Fig. 1 Radial profiles of energetic electron fluxes from 20 February to 06 March 2016. a** Solar wind speed ($V_{sw}$) and north-south component of the interplanetary magnetic field (IMF $B_z$) in Geocentric Solar Magnetospheric (GSM) coordinates (where x-axis points from the Earth to the Sun and the z-axis is the projection of the Earth's dipole axis onto the plane perpendicular to the x axis). **b** Sym-H index (black), which describes the disturbance of the horizontal geomagnetic field component near the equator, and geomagnetic auroral electrojet (AE) index (red), which measures the disturbance of the horizontal component of the geomagnetic field around the auroral oval. **c–e** Daily- and spin-averaged electron fluxes measured by RBSPICE from both Van Allen Probes at 25.6, 39.1, and 62.7 keV, respectively. **f–h** Radial profiles of spin-averaged electron flux during three indicated outbound trajectories of Van Allen Probe A, where the white dashed curves indicate the minimum first-order cyclotron resonant energies of electrons interacting with 24 kHz VLF transmitter waves at the geomagnetic equator (see more details in the section of "Calculation of electron cyclotron resonant energy" in "Methods").

belt at tens of keV. All these waves with weak or moderate amplitudes resonate with electrons to result in predominantly stochastic diffusion processes, which can be evaluated by computing quasi-linear diffusion coefficients and solving the Fokker-Planck diffusion equation[29,30].

**Computations of wave-induced diffusion coefficients.** Figure 3a–l shows the theoretical drift- and bounce-averaged electron pitch-angle diffusion rates at three representative L-shells

calculated using the radial wave amplitude profiles of VLF transmitter waves (red curve in Fig. 2c) and natural magnetospheric waves (Fig. 2d). The details of other input wave parameters including wave frequency spectra, wave normal angle distributions and their latitudinal variations are listed in Supplementary Table 1. While the electron scattering effect of VLF transmitter emissions is weak at $L = 1.8$, it becomes more intense over L-shells of 2.2–2.6 (Fig. 3a–c). At $L = 2.2$ inside the bifurcated energetic electron belt, the loss timescales due to VLF transmitter waves are within a few to tens of days for tens of keV

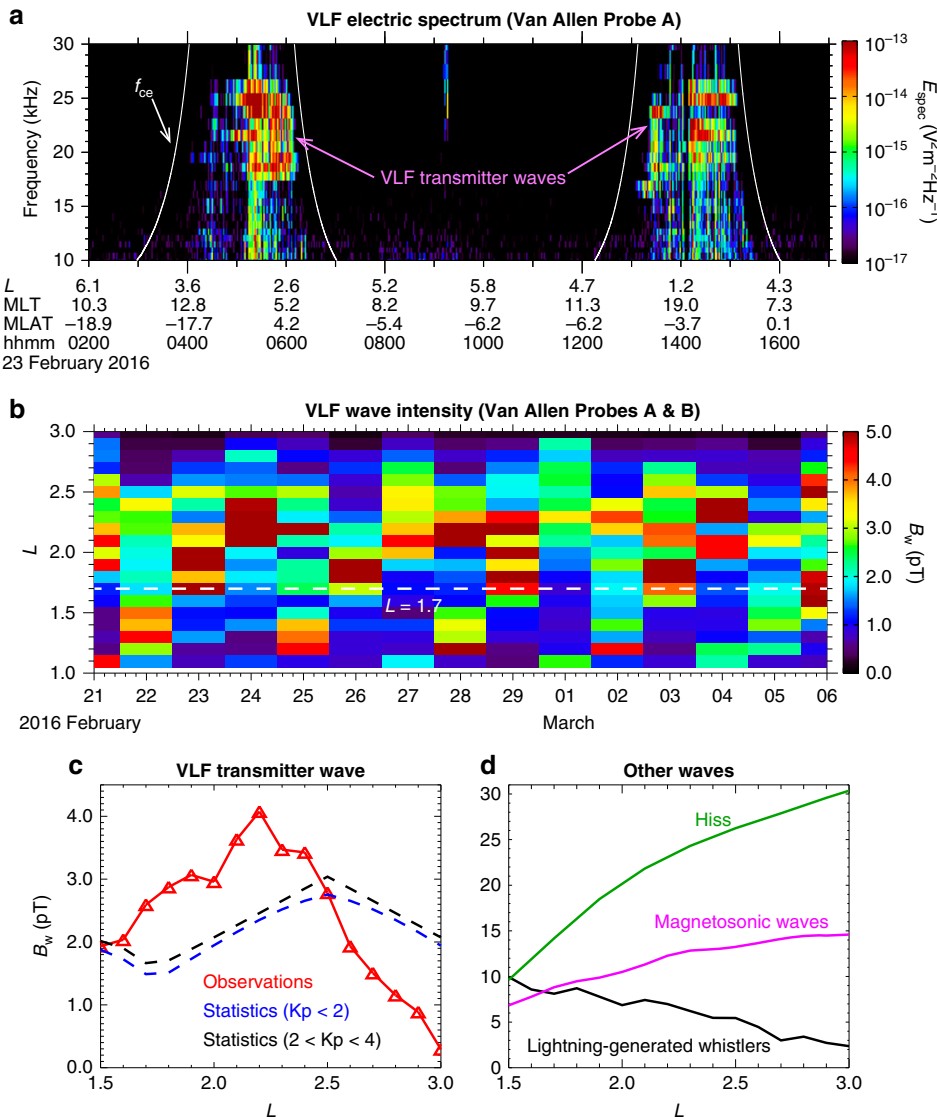

**Fig. 2 In situ observations of VLF transmitter waves and statistical wave amplitudes of concurrent natural magnetospheric waves. a** An example of electric power spectrogram of the high-frequency receiver (HFR) measurements by EMFISIS onboard Van Allen Probe A on 23 February 2016, where the white curves represent the electron cyclotron frequencies ($f_{ce}$). The corresponding $L$-shell, magnetic local time (MLT), and magnetic latitude (MLAT) along the satellite trajectory are also labeled accordingly. **b** The radial profile of daily -averaged magnetic wave amplitude (see more details in the section of "Derivation of various plasma wave amplitudes" in "Methods") of VLF transmitter waves integrated over 10–30 kHz based on the HFR measurements from both Van Allen Probes. Here the horizontal white line represents $L = 1.7$. **c** The radial profile of root-mean-square (RMS) magnetic wave amplitudes of VLF transmitter waves averaged over all MLTs and the entire 15-day period (red line with triangles), and the statistical RMS wave amplitude of VLF transmitter waves under geomagnetically quiet (Kp < 2) and moderate (2 < Kp < 4) conditions[3] using blue and black dashed lines, respectively. Here Kp is a global geomagnetic activity index that is based on 3 h measurements from ground-based magnetometers around the world. **d** Radial profiles of statistical RMS wave amplitudes of lightning-generated whistlers[27] (black line), plasmaspheric hiss[28] (green line) and magnetosonic waves[29] (magenta line) measured by Van Allen Probes under moderate geomagnetic conditions.

electrons (Fig. 3b), as estimated from the inverse of the pitch-angle diffusion coefficients at the bounce loss cone indicated by the white dashed curve. Regarding the natural plasma waves, scattering by plasmaspheric hiss, while much stronger, is primarily effective for electrons above ~100 keV near the bounce loss cone (Fig. 3g–i), and thus plays a dominant role in producing the slot region at higher energies between the inner and outer radiation belts[31–34]. Although lightning-generated whistlers also contribute to the pitch-angle scattering of tens of keV electrons at lower pitch-angles near the bounce loss cone (Fig. 3d–f), their effect is less efficient than that of VLF transmitter waves at $L \leq$ 2.2, since the peak wave power of lightning-generated whistlers occurs at much lower frequencies. Both hiss and lightning-

generated whistlers can scatter tens of keV electrons at pitch angles close to 90° by the Landau resonance, the contribution of which is small to the bifurcation of energetic electrons. The electron scattering rates due to magnetosonic waves are confined to high equatorial pitch-angles and are negligibly small near the bounce loss cone (Fig. 3j–l). Overall, the effects of magnetosonic waves on electrons are weakest compared to the other three types of plasma waves. It is noteworthy that we performed test particle simulations to calculate diffusion coefficients for coherent VLF transmitter waves and plasmaspheric hiss[32,35,36], and found that the test particle simulation results agree well with the quasi-linear calculation results due to the weak wave amplitudes during this relatively quiet event under investigation (see the section of Test

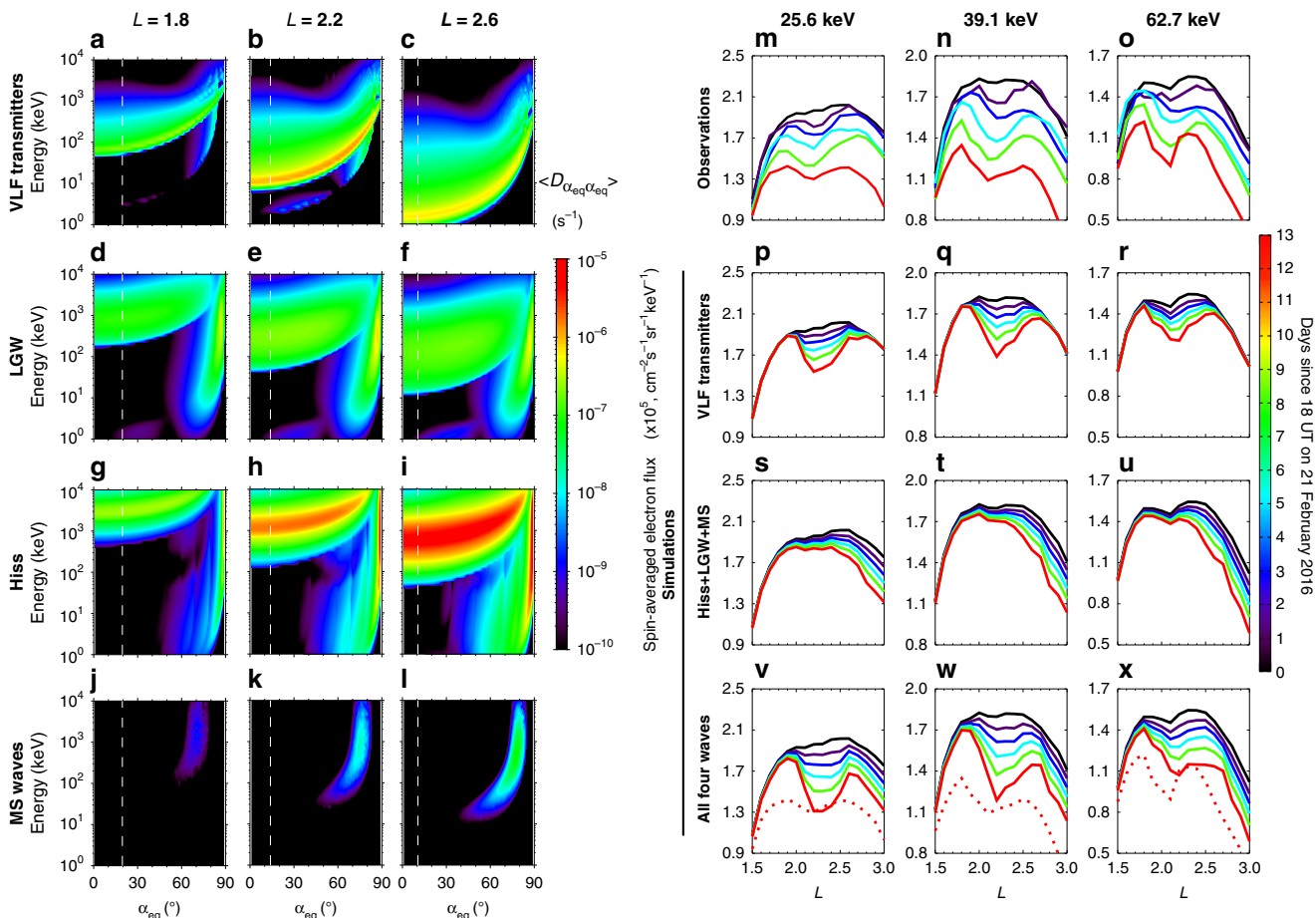

**Fig. 3 Computation of electron diffusion coefficients and comparison of the simulation results to observations. a–c** Drift- and bounce-averaged pitch-angle diffusion coefficients ($\langle D_{\alpha_{eq}\alpha_{eq}} \rangle$) as a function of equatorial pitch-angle ($\alpha_{eq}$) and electron energy at $L = 1.8$, 2.2, and 2.6 for VLF transmitter waves, **d–f**, lightning-generated whistlers (LGWs), **g–i** plasmaspheric hiss, and **j–l** magnetosonic (MS) waves. The vertical white dashed lines in (**a–l**) represent the equatorial bounce loss cone at the given $L$-shell. **m–o** Spatiotemporal evolution of the spin-averaged electron flux profile observed by RBSPICE averaged every 6 h during the period from 21 February to 5 March 2016 for three specific kinetic energies of 25.6, 39.1, and 62.7 keV, corresponding to Fig. 1c–e. Two-dimensional Fokker-Planck simulation results of the temporal evolution of spin-averaged electron fluxes due to three different sets of waves: **p–r** VLF transmitter waves only; **s–u**, combination of plasmaspheric hiss, lightning-generated whistlers (LGWs) and magnetosonic (MS) waves; **v–x** all four-wave types. The red dotted lines in (**v–x**) represent the observed electron fluxes, which are the same as the results of red solid lines in (**m–o**).

particle simulations for coherent plasmaspheric hiss and VLF transmitter waves in the Supplementary Methods and Supplementary Fig. 4).

**Fokker-Planck diffusion simulations**. Using the pitch-angle diffusion coefficients shown in Fig. 3a–l, together with the momentum diffusion and cross-diffusion coefficients (Supplementary Figs. 5 and 6), we simulated the temporal evolution of energetic electrons over $L$-shells of 1.5–3.0 during the 14-day period by numerically solving the two-dimensional Fokker-Planck diffusion equation[29,30]. As this event occurred during a mostly quiet period (Fig. 1a, b), particle injection[37] and radial diffusion[38–40] driven by ultra-low-frequency waves were expected to be insignificant for driving the energetic electron dynamics at low $L$-shells (Supplementary Fig. 7), and thus were not included in the simulations.

The modeling results of the radial profile of energetic electron fluxes under the impact of VLF transmitter waves (Fig. 3p–r) systematically reproduce the key features of the observed evolution of electron belt bifurcation (Fig. 3m–o). A remarkable agreement exists between the observations and simulations in the formation of the local flux minima at $L \sim 2.2$ for tens of keV electrons and the dependence of the bifurcated electron belt on kinetic energy and

$L$-shell. Although lightning-generated whistlers, plasmaspheric hiss, and magnetosonic waves in combination reduce the energetic electron fluxes at $L > \sim2.0$, none of these waves can bifurcate the energetic electron belt (Fig. 3s–u) as VLF transmitter waves do (Fig. 3p–r). The simulation results including both VLF transmitter waves and natural plasma waves (Fig. 3v–x) can reproduce both the main features of the bifurcated electron belt and the magnitudes of energetic electron flux decay, which overall agree well with the observations. These numerical results, together with the simulated evolution of energetic electron spectra over $L$-shells of 1.5–3.0 (Supplementary Fig. 8), provide direct quantitative evidence demonstrating that the bifurcation of the energetic electron belt is predominantly caused by electron scattering due to VLF transmitter waves, aided by electron flux decay mainly due to lightning-generated whistlers and plasmaspheric hiss at $L > \sim2.0$. It is noteworthy that the energetic electron flux decay at $L$-shell of 1.5–1.7 is not well captured by the simulations. At these low $L$-shells, the typical cyclotron resonant electron energies due to VLF transmitter waves are above 100 keV, and losses by an atmospheric collision can potentially play an important role in modulating the electron dynamics[17,41]. Moreover, we performed simulations by using the statistical wave amplitude profile of VLF transmitter waves during the northern hemisphere winter, the results of which

are presented in Supplementary Fig. 9. While the corresponding simulated electron flux decay by adopting the statistical wave amplitudes became slightly slower at $L < 2.3$, it still reproduces the bifurcation of the energetic electron belt at tens of keV.

## Discussion

The bifurcation structure of the energetic electron belt over tens of keV is not unique but is occasionally observed to have a long duration during geomagnetically quiet periods when the effects of VLF transmitters become discernible compared to other naturally driven electron dynamics which strongly depends on geomagnetic activities (see more events in Supplementary Fig. 10). VLF transmitters effectively remove tens of keV electrons to produce a bifurcated energetic electron belt over $L \sim 1.8$–$2.5$ characterized by double radial peaks of electron fluxes, as schematically illustrated in Fig. 4. Our results provide quantitative direct evidence to link operations of VLF transmitters at ground to changes of the energetic electron environment in geospace. Identification of the capability of VLF transmitters to precipitate a considerable portion of energetic electron population over a ~10-day period demonstrates a remarkable feasibility of mitigation of energetic electron fluxes, which is also a major objective of the recently launched Demonstration and Space Experiments (DSX) mission[42]. Moreover, our important findings on energetic electron

loss through pitch-angle scattering driven by plasma waves provide physical insights into understanding fundamental wave-particle interaction processes at the magnetized planets in our solar system and beyond, as well as in active plasma experiments in laboratory and space[42,43].

## Methods

**Calculation of electron cyclotron resonant energy.** The white dashed curves in Fig. 1f–h present the minimum first-order cyclotron resonant energies of electrons (calculated for 0° pitch angle) interacting with 24 kHz VLF waves over $L \sim 1.8$–$2.3$ at the geomagnetic equator. The electron cyclotron resonant energy is calculated by solving both the Doppler-shifted resonance condition[30,44,45],

$$\omega - k_{\parallel}v_{\parallel} = -\frac{N|\Omega_e|}{\gamma} \qquad (1)$$

and the cold plasma dispersion relation[46]

$$An^4 - Bn^2 + C = 0, \qquad (2)$$

with

$$A = S\sin^2\theta + P\cos^2\theta, \qquad (3)$$

$$B = RL\sin^2\theta + PS(1 + \cos^2\theta), \qquad (4)$$

$$C = PRL. \qquad (5)$$

Here $\omega$ is the wave angular frequency, $k_{\parallel}$ and $v_{\parallel}$ are the parallel components of wave number and the electron velocity, respectively, $N$ is the harmonic resonance

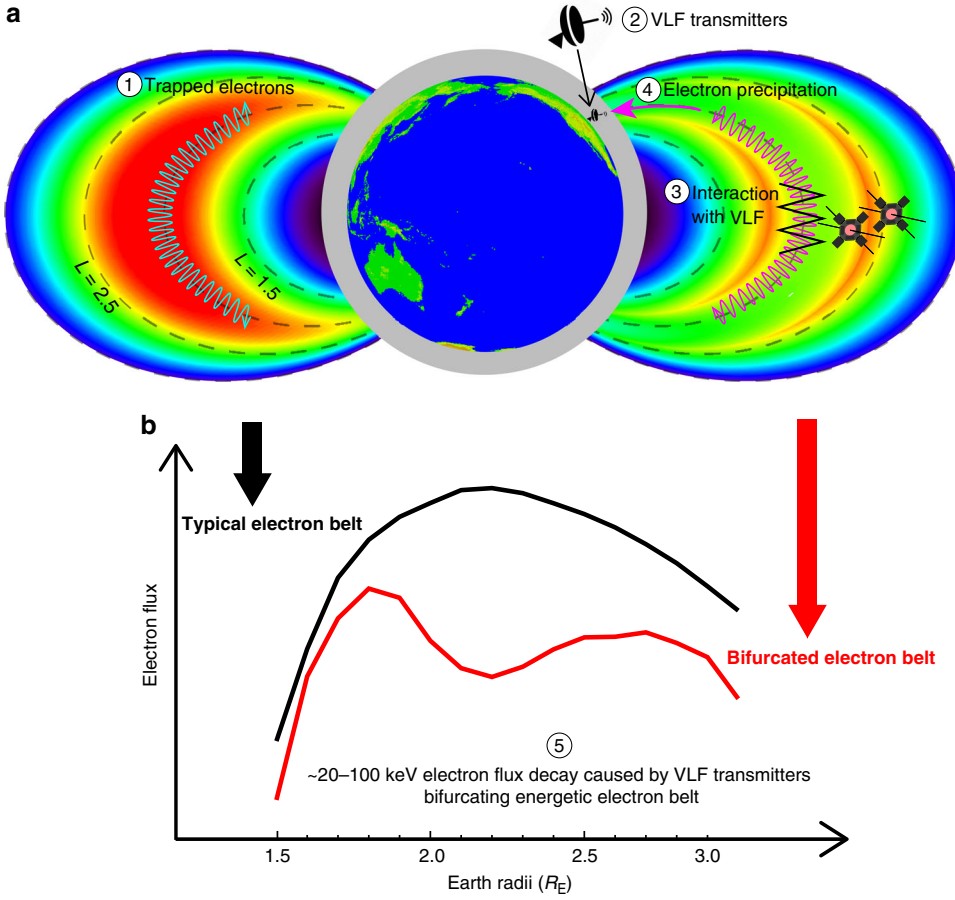

**Fig. 4 Schematic illustration of bifurcating energetic electron belt (tens of keV) caused by VLF transmitters. a** Electron fluxes before (left side) and after (right side) resonant wave-particle interactions with VLF transmitter waves. **b** Variations of radial energetic electron flux profile from a typical (single-peak) structure to a bifurcated (double-peak) belt. There is a multi-step process to directly link the bifurcated electron belt in space to VLF transmitters at ground: ① Electrons are trapped by the geomagnetic field and populate the energetic electron belt over $L \sim 1.5$–$2.5$. ② Ground-based VLF transmitter signals leak into space. ③ VLF transmitter signals interact with counter-streaming energetic electrons along the field line, and ④ drive their precipitation loss into the atmosphere at selected energies. ⑤ Consequently, local electron flux minimum is formed for tens of keV electrons at $L \sim 1.8$–$2.5$ on a timescale of ~10 days, leading to a bifurcated energetic electron belt.

number, $\Omega_e$ is the electron angular gyrofrequency, $\gamma = 1/\sqrt{1 - v^2/c^2}$ is the relativistic Lorentz factor where $v$ and $c$ are the electron speed and light speed, respectively, $n$ is the refractive index, $\theta$ is the wave normal angle, and $R$, $L$, $S$, and $P$ are Stix parameters[46]. The detailed information of the wave normal angle distributions for VLF transmitter waves is described in the section of High-Frequency Receiver (HFR) onboard the Van Allen Probes in the Supplementary Methods and Supplementary Table 1. We adopt the dipole magnetic field model and the empirical plasmaspheric density model[47] to determine the minimum first-order cyclotron resonant energies of electrons at 0° pitch angle.

**Derivation of various plasma wave amplitudes.** The HFR instrument provides only one component of the wave electric field in the plane perpendicular to the spin axis roughly directed toward the Sun. Following previous studies[3,4,48], we multiply this one-component wave spectral intensity by a factor of 3 to obtain the full electric spectral intensity ($I_E$), and subsequently evaluate the magnetic spectral intensity ($I_B$) by applying the Faraday's Law of induction and using the cold plasma dispersion[49],

$$I_B = I_E \left(\frac{n}{c}\right)^2 \sin^2\beta, \tag{6}$$

where $n^2$ is solved using Eqs. (2–5), and $\beta$ is the angle between the wave electric field and the wave propagation vector given by

$$\beta = \arccos\left[\frac{\sin\theta(n^2 - P)}{\sqrt{(n^2\sin^2\theta - P)^2 + \frac{D^2(n^2\sin^2\theta - P)^2}{(n^2 - S)^2} + n^4\sin^2\theta\cos^2\theta}}\right]. \tag{7}$$

Consistent with previous statistical results[3], the wave power of VLF transmitter waves is much stronger on the nightside (00–09 MLT and 15–24 MLT) than on the dayside (09–15 MLT). The computed magnetic wave spectral intensities are then integrated over the frequency range of 10–30 kHz to calculate the magnetic wave amplitudes, which are further averaged over the entire period to acquire the RMS amplitudes of VLF transmitter waves.

Since the statistical wave amplitudes are available as a function of MLT for lightning-generated whistlers, plasmaspheric hiss, and magnetosonic waves, we calculate the RMS wave amplitudes over all MLT sectors to obtain the MLT-averaged wave amplitudes for the three naturally occurring plasma waves[50].

**Calculations of quasi-linear diffusion coefficients.** Because energetic particles bounce between their mirror points along the geomagnetic field line and magnetospheric waves occur over a broad spatial region, the latitudinal variations of the geomagnetic field, plasma density, ion composition, and wave power distribution affect the efficiency of wave-particle interactions. It is necessary to average the local diffusion coefficients to obtain the bounce-averaged diffusion coefficients. The formulae of bounce-averaged diffusion coefficients are given by[51–53]

$$\left\langle D_{\alpha_{eq}\alpha_{eq}}\right\rangle = \frac{1}{S(\alpha_{eq})}\int_0^{\lambda_m} D_{\alpha\alpha}(\alpha)\frac{\cos\alpha\,\cos^7\lambda}{\cos^2\alpha_{eq}}d\lambda, \tag{8}$$

$$\left\langle D_{\alpha_{eq}p}\right\rangle = \left\langle D_{p\alpha_{eq}}\right\rangle = \frac{1}{S(\alpha_{eq})}\int_0^{\lambda_m} D_{\alpha p}(\alpha)\frac{\sin\alpha\,\cos^7\lambda}{\sin\alpha_{eq}\cos\alpha_{eq}}d\lambda \tag{9}$$

$$\left\langle D_{pp}\right\rangle = \frac{1}{S(\alpha_{eq})}\int_0^{\lambda_m} D_{pp}(\alpha)\frac{\sin^2\alpha\,\cos^7\lambda}{\sin^2\alpha_{eq}\cos\alpha}d\lambda \tag{10}$$

where $\lambda_m$ is the mirror latitude of the particle as a function of equatorial pitch-angle, $\alpha$ is the local electron pitch-angle, $\alpha_{eq}$ is the equatorial pitch-angle, and $S(\alpha_{eq})$ is the normalized bounce time with $S(\alpha_{eq}) = 1.30 - 0.56\sin\alpha_{eq}$[54]. $D_{\alpha\alpha}$, $D_{\alpha p}$, and $D_{pp}$ are the local pitch-angle, cross, and momentum diffusion coefficients, and $\left\langle D_{\alpha_{eq}\alpha_{eq}}\right\rangle$, $\left\langle D_{\alpha_{eq}p}\right\rangle$, and $\left\langle D_{pp}\right\rangle$ are the bounce-averaged pitch-angle, cross, and momentum diffusion coefficients.

To quantitatively evaluate the electron scattering effects caused by VLF transmitter waves, lightning-generated whistlers, plasmaspheric hiss, and magnetosonic waves, the Full Diffusion Code[55–58] is implemented to compute the quasi-linear bounce-averaged pitch-angle, momentum, and cross-diffusion coefficients (in unit of s⁻¹) as a function of electron kinetic energy, equatorial pitch-angle, and L-shell. The contributions from resonance harmonics $|N| \leq 10$ (including the Landau resonance) are considered for VLF transmitter waves, lightning-generated whistlers, and plasmaspheric hiss, and the contribution from Landau resonance ($N = 0$) is considered for magnetosonic waves. We further assume that the effects of individual wave modes are additive and independent, which is at least expected to prevail when the diffusion is relatively weak[59,60]. As a result, the diffusion rates by individual wave modes can be summed up to evaluate the combined scattering effect of multiple waves. Besides the bounce-averaged electron pitch-angle diffusion coefficients ($\left\langle D_{\alpha_{eq}\alpha_{eq}}\right\rangle$) shown in Fig. 3a–l for the four-wave modes, Supplementary Figs. 5 and 6 present the corresponding rates of bounce-averaged electron cross diffusion ($\left\langle D_{\alpha_{eq}p}\right\rangle$) and momentum diffusion ($\left\langle D_{pp}\right\rangle$).

**Two-dimensional Fokker-Planck simulation setup.** To simulate the evolution of energetic electron fluxes during the 14-day period, we numerically solve the two-dimensional Fokker-Planck diffusion equation[61–63]

$$\begin{aligned}\frac{\partial f}{\partial t} = &\frac{1}{G}\frac{\partial}{\partial\alpha_{eq}}G\left(\left\langle D_{\alpha_{eq}\alpha_{eq}}\right\rangle\frac{\partial f}{\partial\alpha_{eq}} + p\left\langle D_{\alpha_{eq}p}\right\rangle\frac{\partial f}{\partial p}\right) \\ &+ \frac{1}{G}\frac{\partial}{\partial p}G\left(p\left\langle D_{p\alpha_{eq}}\right\rangle\frac{\partial f}{\partial\alpha_{eq}} + p^2\left\langle D_{pp}\right\rangle\frac{\partial f}{\partial p}\right) - \frac{f}{\tau},\end{aligned} \tag{11}$$

where $f$ is the electron phase space density and related to the differential electron flux $j$ as $f = j/p^2$, $p$ is the electron momentum, $G = p^2 S(\alpha_{eq})\sin\alpha_{eq}\cos\alpha_{eq}$, and $\tau$ equals to a quarter of the bounce period (infinity) inside (outside) the bounce loss cone.

The initial conditions of electron fluxes are collected by RBSPICE onboard Van Allen Probes during 15–21 UT on 21 February when the satellites were near the geomagnetic equator ($|MLAT| < 5°$). For the boundary conditions in the energy ($E$) and equatorial pitch-angle ($\alpha_{eq}$) space adopted for the simulations, the electron phase space density ($f$) keeps constant at $E = 600$ keV, and the RBSPICE electron observations are used to update the PSD distributions at $E = 10$ keV every 6 h. We take $\frac{\partial f}{\partial\alpha_{eq}} = 0$ at $\alpha_{eq} = 0°$, and $\left\langle D_{\alpha_{eq}\alpha_{eq}}\right\rangle\frac{\partial f}{\partial\alpha_{eq}} + p\left\langle D_{\alpha_{eq}p}\right\rangle\frac{\partial f}{\partial p} = 0$ at $\alpha_{eq} = 90°$[62]. Supplementary Fig. 8 shows the quantitative comparisons of the simulated radial profile of energetic electron spectra with the observations at L-shells of 1.5–3.0. It is clearly shown that the combined scattering due to VLF transmitter waves, lightning-generated whistlers, plasmaspheric hiss, and magnetosonic waves can systematically reproduce the key features of the formation and evolution of the bifurcated electron belt. However, the energetic electron flux decay at $L < \sim1.8$ is not well captured by the simulation, which is likely attributed to loss by atmospheric collision[17,41]. Pitch angle scattering by VLF transmitter waves dominates the decay of tens of keV electrons at $L \sim 1.8–2.4$ during the 14-day period.

## Data availability

The particle data adopted for the present study are available from the RBSPICE Data Center (specifically from RBSPICE/Level_3PAP/ESRHELT/v1.1.1) and from the ECT Data Center (specifically from MagEIS/level2/sectors/v8.1.0). The wave and magnetic field data are available from the EMFISIS Data Center (specifically from EMFISIS/L2). Solar wind data and geomagnetic indices are available from OMNIWeb (specifically from OMNI_HRO_1MIN). The source data used to produce figures in the present study are publicly available at https://doi.org/10.6084/m9.figshare.12645206.

## Code availability

The computer code used to simulate the energetic electron evolution due to interactions with plasma waves is available upon request to the corresponding authors.

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

## Acknowledgements

We acknowledge the Van Allen Probes mission, particularly EMFISIS, RBSPICE, and MagEIS teams for providing the wave, magnetic field, and particle data. We also thank the NSSDC OMNIWeb for the use of solar wind and geomagnetic indices data. This work was supported by the B-type Strategic Priority Program of the Chinese Academy of Sciences (Grant No. XDB41000000), the NSFC grants 41674163 and 41474141, the

Chinese Thousand Youth Talents Program, the pre-research projects on Civil Aerospace Technologies No. D020308, D020104, and D020303 funded by the China National Space Administration, the Hubei Province Natural Science Excellent Youth Foundation (2016CFA044), the NSF grants of AGS-1723588 and AGS-1847818, and the Alfred P. Sloan Research Fellowship FG-2018-10936.

## Author contributions

M.H. performed the data analysis and numerical simulations, produced the original figures, and wrote the initial draft of the manuscript. W.L. and B.N. led the study, supervised the project development at Boston University and Wuhan University, respectively, and contributed significantly to explain the results, and finalize the figures and the manuscript. Q.M. initialized the concept of VLF transmitter wave scattering, provided statistical models of VLF transmitter waves and magnetosonic waves, helped with data analysis, and contributed to the writing of the manuscript. A.G. and X.S. provided the statistical wave model of lightning-generated whistlers. S.G.C., J.B., X.G., S.F., Z.X., and G.D.R. helped analyze the observational and simulation results and contributed to the writing of the manuscript through reviews and edits.

## Competing interests

The authors declare no competing interests.
