## [Peer Review File · Nature Communications]

REVIEWER COMMENTS

Reviewer #1 (Remarks to the Author):

This manuscript reports the correspondence between observed in situ 24 kHz VLF transmissions from the terrestrial transmitter NAA (Cutler, Maine) with the formation of a radially bifurcated trapped inner zone electron radiation belt, as observed by the Van Allen Probes. In addition, simulations of belt dynamics using quasi-linear theory were performed that largely reproduce the observed structure.

These results are interesting and new. Although quasi-linear theory has been successfully used to understand the structure of the trapped electron belts in the past, its application to this bifurcated configuration of the inner zone electrons is novel. This is in part because the observed structure is difficult to catch, requiring days of geomagnetically quiet conditions so as not to be washed out by other processes. The outcome of the study is particularly interesting because it reinforces the weak influence of most of the terrestrial VLF transmitters on radiation belt electrons – an understanding that is beginning to emerge since the early estimates of the relative weights of such transmitters, lightning-generated whistlers, and plasmaspheric hiss.

The paper is clearly written and the figures are both appropriate and convey the results effectively. The reference list is largely comprehensive and supports the methodology and conclusions. The data used are available and the analysis technique is well-described, building on much significant past work.

No major revisions are required prior to publication. It is my judgment that this is a very well-written, clear, impactful manuscript that provides important new insights to the field.

Some specific comments:

In the section reviewing past connections of terrestrial transmitters with precipitating electrons, the authors may wish to consider referring to Graf, et al., 2009 (doi 10.1029/2008JA013949) and references therein. Although that work did not provide clear quantitative evidence of the order seen in this manuscript, the authors did estimate pitch angle diffusion in an attempt to perform a similar comparison.

On line 102, it is observed that rms wave magnetic field amplitudes from the VLF transmitters significantly exceed the statistical values. Given the reference cited for the statistical values, it seems likely that the measurements and processing approaches are the same for both sets of curves. This suggests that the differences between the two are primarily the result of seasonal influences. The reader is left to wonder, however, and this might be worth a few words to clarify.

On line 132, there is a missing reference. I presume this is reference 26.

On line 169, it is stated that the Air Force's DSX spacecraft has a major objective of testing mitigation of energetic electron fluxes. The Air Force has never stated this publicly to my knowledge, but a recent press release does in fact confirm this: <https://www.kirtland.af.mil/News/Article-Display/Article/2187232/afri-satellite-duo-probing-earths-radiation-belts/>

In the section from lines 194 to 212, researchers familiar with these types of analyses will immediately be concerned about the choices for the wave normal angles, as this has a substantial impact on the outcomes of the calculations. Although the choices are well-supported by the literature, a pointer in this section to where this is addressed in the Supplementary Material would be appropriate.

Reviewer #2 (Remarks to the Author):

Evaluation:

This paper examines the effects of VLF transmitters on energetic electrons in the region $1.5 < L < 3$. The authors show that the observed bifurcation of the radial distribution of energetic electrons, with energies in the range $20 < E < 100$ keV, in the region $1.8 < L < 2.4$ can only be explained by wave particle interactions with VLF transmitters. This is a very interesting and novel result which will be interesting to others in the community and the wider field. I find the work convincing but would strongly recommend that the authors consider using their statistical values rather than values derived from the in situ measurements for the VLF transmitter rms wave magnetic amplitudes (see comment below). In summary, this is a very interesting study, which merits publication in Nature Communications. However, I have a number of comments and suggestions that I would like the authors to consider before I can recommend publication.

Comments

Lines 22 and 47. There is an important paper from 2008 that provides direct evidence of the effects of VLF transmitter waves on energetic electrons (Gamble et al., 2008), which should be cited. In this paper the authors report on enhancements in drift loss cone electron fluxes at an altitude of 800 km by the DEMETER spacecraft. These enhancements are found at $1.6 \lesssim L \lesssim 2$ and $100 \lesssim E \lesssim 375$ keV. Longitudinally, they are observed eastward of the Australian transmitter NWC (114°E) but westward of the South Atlantic Anomaly. They are a result of wave-particle interactions with VLF transmitter waves from NWC, scattering electrons into the drift loss cone, before they are lost to the atmosphere on the western side of the SAA.

Reference

Gamble, R. J., Rodger, C. J., Clilverd, M. A., Sauvaud, J. A., Thomson, N. R., Stewart, S. L., et al. (2008). Radiation belt electron precipitation by man-made VLF transmissions. *Journal of Geophysical Research*, 113, A10211. <https://doi.org/10.1029/2008JA013369>

Line 29. Loss timescales for tens of keV electrons due to VLF transmitters are known to fall into the region of 10s of days in the inner slot region (e.g., Ross et al., 2019). I would therefore remove the word 'remarkable' from this statement.

Reference

Ross, J. P. J., Meredith, N. P., Glauert, S. A., Horne, R. B., & Clilverd, M. A (2019). Effects of VLF transmitter waves on the inner belt and slot region. *Journal of Geophysical Research: Space Physics*, 124. <https://doi.org/10.1029/2019JA026716>

Line 42. What do you mean by a 'dense' population?

Line 50. 'observed characteristically' should be 'observed to be characteristically'

Line 88. Can you please explain the 'first order cyclotron resonant energy' in more detail. Is this the minimum resonant energy associated with an electron with zero velocity perpendicular to the ambient magnetic field ?

Line 101. While it may be of interest to plot the radial profile of the averaged observed wave magnetic field amplitudes, the spatial coverage in this relatively short time period will not give the best measure of the average wave magnetic field intensities experienced by the energetic electrons as they drift around the Earth. The wave intensity experienced by an electron at any given time will depend on both the magnetic local time and the geographic longitude. The best way to do this is to build a comprehensive map using data from multiple years of satellite data. I would therefore strongly recommend that the authors consider using the statistical values as opposed to the in situ values. Inspection of Figure 2c suggests that this will increase the loss timescales out to $L = \sim 2.5$ but reduce them further out. This would reduce the efficiency of the loss process in the region $L = 1.8-$

2.5 but would help remove some of the energetic particles that the simulations are unable to currently remove at higher L shells as time progresses (Figure S5).

Figure 3.

To help compare the modelling results with the data and to facilitate the interpretation of the results I would recommend plotting the final observed profiles (red lines on Figures 3m, 3n and 3o) as dotted red lines on the simulations in Figures 3v, 3w and 3x.

Reviewer #3 (Remarks to the Author):

Review of “Human-Made Very Low-Frequency Transmitters Bifurcate Energetic Electron Belt in Near-Earth Space” by Man, Li, Ni, Ma, Green, Claudepierre, Bortnik, Giu, Fu, Xiang and Reeves

The paper shows very interesting data, but I would say that the claim that you provide “compelling quantitative evidence that VLF transmitter waves are causing the energetic electron bifurcation” is a bit of an overstatement. I think more work needs to be done on this topic to make this acceptable for publication in Nature Comm.

Major Comments

The approach that the authors use, Fokker Planck diffusion simulations is dated. Recent new work on plasmaspheric hiss (JGRSP, 120, 414-431, doi:10.1002/2014JA020518, 2015; JGRSP, 122, 1643–1657, doi:10.1002/2016JA023289, 2017; JGRSP, 123, <https://doi.org/10.1029/2018JA025975>, 2018; JGRSP, 124, 10063-10084,

<https://doi.org/10.1029/2019JA027102>, 2019) have shown that plasmaspheric hiss is intense and coherent. With coherent waves, the loss rate is ~two orders of magnitude faster. The authors should do the simulations/calculations assuming coherent hiss waves (see JGR, 115, A00F15, doi:10.1029/2009JA014885, 2010 for chorus wave-particle interactions) for a more accurate comparison with the VLF transmitter waves. A question for the authors: “Are the VLF transmitter waves coherent in the area that the wave-particle interactions are taking place?” “Have the authors considered the possibility that magnetosonic waves are coherent, and if so, how will wave-particle interactions be changed?”

The electron energy being examined should be mentioned in the title of the paper. Most people think of the “electron gap” as being due to 100s of keV electrons, not these lower energy electrons. This should also be mentioned more prominently in the body of the paper. The higher energy electron slot is due to the combination of coherent (?) magnetosonic waves and coherent hiss? Some crude approximations are given in the 2019 paper above.

Minor Comments

Lines 39-40. What does “when a magnetic field line crosses the geomagnetic equator” mean and how does this geometry affect transmitter waves allowing them to penetrate through the ionosphere? It would be good for the reader if you would elaborate a bit more and give references.

Lines 54-56. The energy of ~30 to 200 keV is beyond the limit of your study, so how is this relevant?

Lines 76-77. Moderate substorms. Chorus is generated during substorms. It has been hypothesized that chorus propagates into the plasmasphere. This seems like a reasonable competing mechanism to VLF transmitters?

Lines 102-109. I don't understand the discussion here. If these natural waves (lightning, hiss and magnetosonic waves) “significantly exceed the VLF transmitter intensity”, then why do the simulations indicate that it is transmitter signals that are causing the slot? Perhaps what you mean is that at VLF frequencies the transmitter intensities have the highest intensities? Or have the highest intensities in the VLF frequency range? Something seems to be missing here.

Lines 119-121. Here you mention hiss is responsible for the high energy electron slot. Please discuss the possibility of either hiss Landau interactions with the 10s of keV electrons or the high frequency ends of hiss for cyclotron resonance. This is a bit confusing.

Lines 124-126. Previous authors have discussed the combination of magnetosonic waves with plasmaspheric hiss for the pitch angle diffusion of energetic electrons, not magnetosonic waves by themselves. So this statement is a bit unfair.

Line 132. Radial diffusion. Here are a couple of papers on this topic: GRL., 26, 3273, 1999; SW, 2, S10S02, doi:10.1029/2004SW000070., 2004. However this mechanism is usually quoted for relativistic electrons, not the low energies that you are discussing. Could low energy electrons radially diffuse much by this process?

We thank all reviewers for careful reading of the manuscript and valuable and constructive comments to improve the quality of the paper. We have made point-by-point responses to the detailed comments by all reviewers and incorporated the suggested changes in the revised manuscript accordingly, as highlighted in the blue color.

Responses to Reviewer #1:

This manuscript reports the correspondence between observed in situ 24 kHz VLF transmissions from the terrestrial transmitter NAA (Cutler, Maine) with the formation of a radially bifurcated trapped inner zone electron radiation belt, as observed by the Van Allen Probes. In addition, simulations of belt dynamics using quasi-linear theory were performed that largely reproduce the observed structure.

These results are interesting and new. Although quasi-linear theory has been successfully used to understand the structure of the trapped electron belts in the past, its application to this bifurcated configuration of the inner zone electrons is novel. This is in part because the observed structure is difficult to catch, requiring days of geomagnetically quiet conditions so as not to be washed out by other processes. The outcome of the study is particularly interesting because it reinforces the weak influence of most of the terrestrial VLF transmitters on radiation belt electrons – an understanding that is beginning to emerge since the early estimates of the relative weights of such transmitters, lightning-generated whistlers, and plasmaspheric hiss.

The paper is clearly written and the figures are both appropriate and convey the results effectively. The reference list is largely comprehensive and supports the methodology and conclusions. The data used are available and the analysis technique is well-described, building on much significant past work.

No major revisions are required prior to publication. It is my judgment that this is a very well-written, clear, impactful manuscript that provides important new insights to the field.

In the section reviewing past connections of terrestrial transmitters with precipitating electrons, the authors may wish to consider referring to Graf, et al., 2009 (doi 10.1029 2008JA013949) and references therein. Although that work did not provide clear quantitative evidence of the order seen in this manuscript, the authors did estimate pitch angle diffusion in an attempt to perform a similar comparison.

Reply:

We thank the reviewer for this valuable comment. We have added related citations in the main text of the revised manuscript as follows: “Early studies provided evidence of the potential transmitter-induced electron precipitation by correlating the electron flux enhancement inside the drift-loss cone with the VLF wave power¹²⁻¹⁴”. Please see Lines 47 – 49.

Reference:

12. Sauvaud, J.- A. et al. Radiation belt electron precipitation due to VLF transmitters: Satellite observations, *Geophys. Res. Lett.*, **35**, L09101 (2008).
13. Gamble, R. J. et al. Radiation belt electron precipitation by man- made VLF transmissions, *J. Geophys. Res.*, **113**, A10211 (2008).
14. Graf, K. L. et al. DEMETER observations of transmitter- induced precipitation of inner radiation belt electrons, *J. Geophys. Res.*, **114**, A07205 (2009).

On line 102, it is observed that rms wave magnetic field amplitudes from the VLF transmitters significantly exceed the statistical values. Given the reference cited for the statistical values, it seems likely that the measurements and processing approaches are the same for both sets of curves. This suggests that the differences between the two are primarily the result of seasonal influences. The reader is left to wonder, however, and this might be worth a few words to clarify.

Reply:

We thank the reviewer for this constructive suggestion. We agree with the reviewer that the difference of the wave amplitude of VLF transmitter waves between the *in situ* measurements and statistical results is primarily due to the seasonal influences. The wave power at $L > 1.7$ mainly comes from the NAA and NLK transmitters located in North America, where the transionospheric wave attenuation decreases due to a lower sunlit electron density in February, so that the wave amplitude during February at $L > 1.7$ is stronger than the wave amplitude averaged during all seasons. We have added this point to the main text and Supplementary Information. Please see Lines 109 – 114 in the main text and Lines 203–207 in the Supplementary Information (including the new Figure S2).

On line 132, there is a missing reference. I presume this is reference 26.

Reply:

We thank the reviewer for pointing this out. This reference is Hudson et al. (2008) and we have added this reference number in the main text of the revised manuscript. Please see Line 158.

Reference:

Hudson, M. K., Kress, B. T., Mueller, H. R., Zastrow, J. A., & Blake, J. B. Relationship of the Van Allen radiation belts to solar wind drivers. *J. Atmos. Sol.-Terr. Phys.* **70**, 708-729 (2008).

On line 169, it is stated that the Air Force's DSX spacecraft has a major objective of testing mitigation of energetic electron fluxes. The Air Force has never stated this publicly to my knowledge, but a recent press release does in fact confirm this: <https://www.kirtland.af.mil/News/Article-Display/Article/2187232/afri-satellite-duo-probing-earths-radiation-belts/>

Reply:

We thank the reviewer for introducing this recent press release article. The DSX

mission has three physics-based research/experiment areas, one of which is Wave Particle Interaction Experiment (WPIx): Researching the physics of VLF electromagnetic wave transmissions through the ionosphere and in the magnetosphere and characterizing the feasibility of natural and man-made VLF waves to reduce and precipitate space radiation (Scherbarth et al., 2009). Therefore, we think the reference of Scherbarth et al. (2009) supports this objective of DSX mission.

Reference:

Scherbarth, M., Smith, D., Adler, A., Stuart, J. & Ginet, G. AFRL's Demonstration and Science Experiments (DSX) mission. *Proc. SPIE Soc. Opt. Eng.* **7438**, 74380B (2009).

In the section from lines 194 to 212, researchers familiar with these types of analyses will immediately be concerned about the choices for the wave normal angles, as this has a substantial impact on the outcomes of the calculations. Although the choices are well-supported by the literature, a pointer in this section to where this is addressed in the Supplementary Material would be appropriate.

Reply:

We thank the reviewer for this valuable comment. We have added this point to the main text of the revised manuscript as follows. "The detailed information of the wave normal angle distributions for VLF transmitter waves is described in Section 2 and Table S1 in the Supplementary Information." Please see Lines 221 – 223.

Responses to Reviewer #2:

This paper examines the effects of VLF transmitters on energetic electrons in the region $1.5 < L < 3$. The authors show that the observed bifurcation of the radial distribution of energetic electrons, with energies in the range $20 < E < 100$ keV, in the region $1.8 < L < 2.4$ can only be explained by wave particle interactions with VLF transmitters. This is a very interesting and novel result which will be interesting to others in the community and the wider field. I find the work convincing but would strongly recommend that the authors consider using their statistical values rather than values derived from the in situ measurements for the VLF transmitter rms wave magnetic amplitudes (see comment below). In summary, this is a very interesting study, which merits publication in Nature Communications. However, I have a number of comments and suggestions that I would like the authors to consider before I can recommend publication.

Lines 22 and 47. There is an important paper from 2008 that provides direct evidence of the effects of VLF transmitter waves on energetic electrons (Gamble et al., 2008), which should be cited. In this paper the authors report on enhancements in drift loss cone electron fluxes at an altitude of 800 km by the DEMETER spacecraft. These enhancements are found at $1.6 \leq L \leq 2$ and $100 \leq E \leq 375$ keV. Longitudinally, they are observed eastward of the Australian transmitter NWC ($114^\circ E$) but westward of the South Atlantic Anomaly. They are a result of wave-particle interactions with VLF transmitter waves from NWC, scattering electrons into the drift loss cone, before they are lost to the atmosphere on the western side of the SAA.

Reference

*Gamble, R. J., Rodger, C. J., Clilverd, M. A., Sauvaud, J. A., Thomson, N. R., Stewart, S. L., et al. (2008). Radiation belt electron precipitation by man-made VLF transmissions. *Journal of Geophysical Research*, **113**, A10211. <https://doi.org/10.1029/2008JA013369>*

Reply:

We thank the reviewer for this valuable comment. We have added related citations in the main text of the revised manuscript as follows. “Early studies provided evidence of the potential transmitter-induced electron precipitations by correlating electron flux enhancement inside the drift-loss cone and VLF wave power¹²⁻¹⁴”. Please see Lines 47 – 49.

Reference:

- 12. Sauvaud, J.- A. et al. Radiation belt electron precipitation due to VLF transmitters: Satellite observations, *Geophys. Res. Lett.*, **35**, L09101 (2008).*
- 13. Gamble, R. J. et al. Radiation belt electron precipitation by man- made VLF transmissions, *J. Geophys. Res.*, **113**, A10211 (2008).*
- 14. Graf, K. L. et al. DEMETER observations of transmitter- induced precipitation of inner radiation belt electrons, *J. Geophys. Res.*, **114**, A07205 (2009).*

Line 29. Loss timescales for tens of keV electrons due to VLF transmitters are known to fall into the region of 10s of days in the inner slot region (e.g., Ross et al., 2019). I

would therefore remove the word 'remarkable' from this statement.

Reference

Ross, J. P. J., Meredith, N. P., Glauert, S. A., Horne, R. B., & Clilverd, M. A (2019). Effects of VLF transmitter waves on the inner belt and slot region. *Journal of Geophysical Research: Space Physics*, 124. <https://doi.org/10.1029/2019JA026716>

Reply:

We thank the reviewer for pointing this out. We have deleted the word "remarkable". Please see Line 30.

Line 42. What do you mean by a 'dense' population?

Reply:

To avoid confusion, we deleted "dense" here. Please see Line 44.

Line 50. 'observed characteristically' should be 'observed to be characteristically'

Reply:

We thank the reviewer for pointing this out. We have changed 'observed characteristically' to 'observed to be characteristically'. Please see Line 53.

Line 88. Can you please explain the 'first order cyclotron resonant energy' in more detail? Is this the minimum resonant energy associated with an electron with zero velocity perpendicular to the ambient magnetic field?

Reply:

Yes, the reviewer is correct that we calculated the minimum first-order cyclotron resonant energy for electrons with 0° pitch angle at the geomagnetic equator. The detailed information of the first-order cyclotron resonant energy calculation is described in the "Methods" section, and we have added the information regarding minimum first-order cyclotron resonant energies of electrons calculated for 0° pitch angle in Lines 206, and 224–225.

Line 101. While it may be of interest to plot the radial profile of the averaged observed wave magnetic field amplitudes, the spatial coverage in this relatively short time period will not give the best measure of the average wave magnetic field intensities experienced by the energetic electrons as they drift around the Earth. The wave intensity experienced by an electron at any given time will depend on both the magnetic local time and the geographic longitude. The best way to do this is to build a comprehensive map using data from multiple years of satellite data. I would therefore strongly recommend that the authors consider using the statistical values as opposed to the in situ values. Inspection of Figure 2c suggests that this will increase the loss timescales out to $L = \sim 2.5$ but reduce them further out. This would reduce the efficiency of the loss process in the region $L = 1.8-2.5$ but would help remove some of the energetic particles that the simulations are unable to currently remove at higher L shells as time progresses (Figure S5).

Reply:

We thank the reviewer for this valuable comment. We agree with the reviewer that it

is also of great importance to perform simulations using the statistical wave amplitude of VLF transmitter waves. Previous statistical results suggest that the wave amplitude distributions of VLF transmitter waves depend on seasons (e.g., Ma et al., 2017). The wave power at $L > 1.7$ mainly comes from the NAA and NLK transmitters located in North America, where the transionospheric wave attenuation decreases with a lower sunlit electron density, so that the VLF wave amplitudes during February at $L > 1.7$ are stronger than the wave amplitudes averaged during all seasons. The comparison of the VLF transmitter wave amplitudes between *in situ* observation and statistical values (Ma et al., 2017) during different seasons is added in Fig. S2 in Supplementary Information. Clearly, the wave amplitude of VLF transmitters is stronger at $L > 1.7$ during northern hemisphere winter than summer. Overall, the statistical wave amplitudes during northern hemisphere winter are weaker than the *in situ* observed values at $L < 2.3$, while they exceed the observed values at $L > 2.5$.

Following the reviewer's suggestion, we adopt the statistical wave amplitude during northern hemisphere winter to perform all the diffusion simulations from 21 February to 06 March 2016, the results of which are shown in Fig. S9 in the same format as those in Fig. 3 in the main text. The simulated results of the energetic electron flux radial profile by only including VLF transmitter waves (Fig. S9p-r) reproduce the key features of the observed evolution of electron belt bifurcation, which is similar to the simulated results using *in situ* observed wave amplitude of VLF transmitter waves, although the flux decay is slightly slower at $L < 2.3$ due to the weaker wave power from the statistical results. By including both human-made and natural plasma waves (Fig. S9v-x), the simulated results can reproduce the main features of the bifurcated electron belt at 25.6 keV and 39.1 keV, while the results for 62.7 keV do not show a clear feature of bifurcation as the results shown in Fig. 3, which is mainly due to the weaker wave amplitude from the statistical results at $L < 2.3$. Overall, the simulation results using statistical wave amplitudes of VLF transmitter waves demonstrate a similar trend to the results using *in situ* observed values.

Besides, previous statistical results demonstrate that the magnetic local time coverage of VLF transmitter waves along the electron drift trajectory is about 50% due to the reason that VLF transmitter waves are considerably stronger on the nightside than the dayside (Ma et al., 2017). In the studied event, the Van Allen Probes traveled over ~1–6 MLT during outbound trajectories over L shells of 1.5–3.0, where the VLF transmitter signals are more likely to be observed and stronger, while the satellites traveled over ~13 – ~18 MLT during inbound trajectories over L shells of 1.5–3.0, where the VLF transmitter waves are less likely to be observed and weaker. In spite of the limited time of 15-day observations during this event, the averaged *in situ* observed wave amplitude during this event can provide a reasonable estimation of the MLT-averaged wave amplitude.

We have added this point to the main text and Supplementary Information. Please see Lines 180 – 184 in the main text, and Fig. S2 & S9 in the Supplementary Information.

Figure 3.

To help compare the modelling results with the data and to facilitate the interpretation of the results I would recommend plotting the final observed profiles (red lines on Figures 3m, 3n and 3o) as dotted red lines on the simulations in Figures 3v, 3w and 3x.

Reply:

We thank the reviewer for this valuable comment. We have added the observed profiles as red dotted lines on the simulations in Figs. 3v, 3w, and 3x. Please see the revised Fig. 3 and the corresponding figure caption in the main text.

Responses to Reviewer #3:

Review of “Human-Made Very Low-Frequency Transmitters Bifurcate Energetic Electron Belt in Near-Earth Space” by Man, Li, Ni, Ma, Green, Claudepierre, Bortnik, Gu, Fu, Xiang and Reeves

The paper shows very interesting data, but I would say that the claim that you provide “compelling quantitative evidence that VLF transmitter waves are causing the energetic electron bifurcation” is a bit of an overstatement. I think more work needs to be done on this topic to make this acceptable for publication in Nature Comm.

Major Comments

The approach that the authors use, Fokker Planck diffusion simulations is dated. Recent new work on plasmaspheric hiss (JGRSP, 120, 414-431, doi:10.1002/2014JA020518, 2015; JGRSP, 122, 1643-1657, doi:10.1002/2016JA023289, 2017; JGRSP, 123, https://doi.org/10.1029/2018JA025975, 2018; JGRSP, 124, 10063-10084, https://doi.org/10.1029/2019JA027102, 2019) have shown that plasmaspheric hiss is intense and coherent. With coherent waves, the loss rate is ~two orders of magnitude faster. The authors should do the simulations/calculations assuming coherent hiss waves (see JGR, 115, A00F15, doi:10.1029/2009JA014885, 2010 for chorus wave-particle interactions) for a more accurate comparison with the VLF transmitter waves. A question for the authors: “Are the VLF transmitter waves coherent in the area that the wave-particle interactions are taking place?” “Have the authors considered the possibility that magnetosonic waves are coherent, and if so, how will wave-particle interactions be changed?”

Reply:

We thank the reviewer for introducing several interesting papers and the valuable comment regarding the coherency of plasma waves. We cited these references accordingly and added them into the reference list (Lines 148–150). Since the frequency range of VLF transmitter waves is typically very narrow (see an example in Fig. S3 in the Supplementary Information), they are reasonable to be considered as coherent waves. We performed test particle simulations to evaluate the effect of coherent hiss and VLF transmitter waves and included the detailed results in the Supplementary Information (Section 4 and Fig. S4).

The test particle simulations are performed to evaluate the electron scattering effects by coherent hiss and VLF transmitter waves at $L = 2.3$, which is a typical L -shell where the observed fluxes decreased significantly during this event, and compared to the quasi-linear theory results. We assume that these two types of plasma waves have a single wave frequency (to represent the most coherent wave), which is adopted from the central wave frequency from the statistical wave frequency spectra (Li et al., 2015; Ma et al., 2017). For simplicity, we assume the wave normal angle (θ) for hiss and VLF transmitter waves as $\theta = 0^\circ$. The simulations numerically solve the full electron momentum equation given below (Bell, 1984; Bortnik et al., 2008; Li, J. et al., 2015):

$$\frac{d\vec{p}}{dt} = q_e \left(\vec{E}_w + \frac{\vec{p}}{m_e \gamma} \times (\vec{B}_w + \vec{B}_0) \right), \quad (\text{R1})$$

where \vec{p} and q_e are the electron momentum and charge, and $\gamma = (1 - v^2/c^2)^{-1/2}$ is the relativistic Lorentz factor where v and c are the electron speed and light speed, respectively. \vec{E}_w and \vec{B}_w are the wave electric and magnetic field, and \vec{B}_0 is the background magnetic field. The momentum equation (Eq. R1) can be rewritten as a set of three gyro-averaged ordinary differential equations (Bell, 1984; Bortnik, 2004; Bortnik et al., 2008; Li, J. et al., 2015), which are then numerically solved to perform the test particle simulations. Same as in quasi-linear calculations, we assume that hiss waves cover the latitudinal range of $|\text{MLAT}| \leq 45^\circ$, while VLF transmitters can reach the magnetic latitude where the magnetic field line reaches the altitude of 800 km from the Earth's surface, which is $|\text{MLAT}| \leq 45.61^\circ$ at $L = 2.3$. Plasma waves are launched from the equator and propagate to higher magnetic latitudes until the maximum magnetic latitude (λ_{max}) in the northern hemisphere. Energetic electrons are released at latitude of the lower value between λ_{max} and the mirror point latitude in the northern hemisphere and move towards the equator. We trace the electrons until they reach the equator for the first time. For each initial energy and pitch angle, 72 electrons are released with the initial phase uniformly distributed between 0° and 360° . The wave amplitude of hiss is based on statistical results from Van Allen Probes measurement (Li et al., 2015), ~ 24.3 pT at $L = 2.3$, while the wave amplitude of VLF transmitter waves is based on *in situ* satellite measurements, ~ 3.47 pT at $L = 2.3$, the same as the simulations in the main text. The detailed input wave parameters for test particle simulations including wave frequency, wave normal angle, wave latitudinal variations, and wave amplitude are listed in Table S2 in the Supplementary Information. For comparison, we also calculate quasi-linear diffusion coefficients by adopting similar parameters used in the test particle simulations, which are also listed in Table S2. For field-aligned electromagnetic waves, only the first order cyclotron resonance ($N = -1$ for R-mode) contributes to the electron scattering effect; therefore, the resonance harmonic $N = -1$ was included in the quasi-linear calculations for both hiss and VLF transmitter waves.

Fig. S4 presents the comparison of bounce-averaged pitch-angle diffusion coefficients due to plasmaspheric hiss and VLF transmitter waves calculated by using test particle simulations (**a-b**) with quasi-linear calculations (**c-d**). Overall, the results of the test particle simulations agree well with the quasi-linear results, indicating that the wave amplitudes of plasmaspheric hiss and VLF transmitter waves are not sufficiently strong (< 25 pT) to cause a significant nonlinear effect. Therefore, the coherent hiss or VLF transmitter waves do not significantly affect our quasi-linear simulation results. Our result is also consistent with the previous study (Tao et al., 2012), which indicates that bounce-averaged quasi-linear diffusion coefficients are still valid for narrowband whistler mode waves, as long as the amplitude is small ($< a$ few hundred pT).

Furthermore, the study of Li et al. (2014, <https://doi.org/10.1002/2014GL060461>) has

performed the comparisons of pitch-angle diffusion coefficients for magnetosonic waves using both quasi-linear calculations and test particle simulations, which demonstrates good consistency between the results from test particle simulations and quasi-linear calculations in the high-density plasmasphere, where the transit-time effect is not very prominent. Besides, since the diffusion coefficients of magnetosonic waves (shown in Fig. 1 and Figs. S2-S3) are the weakest among all these four types of plasma waves, their contributions to electron flux decay is very minor compared to plasmaspheric hiss, lightning-generated whistlers, and VLF transmitter waves.

In conclusion, due to the reason that the wave amplitudes of plasmaspheric hiss, VLF transmitter waves, and magnetosonic waves are not strong enough ($< \sim 25$ pT) to cause significant nonlinear effects during this relatively quiet event, and the diffusion coefficients calculated by performing test particle simulations are similar to those using the quasi-linear approach. Therefore, we think it is reasonable to evaluate electron scattering effects due to plasmaspheric hiss, lightning-generated whistlers, VLF transmitter waves, and magnetosonic waves by performing Fokker-Planck diffusion simulations. We have added these points to the revised manuscript in the main text and Supplementary Information. Please see Lines 147 – 152 in the main text and Lines 111 – 150 in the Supplementary Information.

Reference:

- Bell, T. F. The nonlinear gyroresonance interaction between energetic electrons and coherent VLF waves propagating at an arbitrary angle with respect to the Earth's magnetic field. *J. Geophys. Res.* **89**, 905–918 (1984).
- Bortnik, J., Thorne, R. M., & Inan, U. S. Nonlinear interaction of energetic electrons with large amplitude chorus. *Geophys. Res. Lett.* **35**, L21102 (2008).
- Falkowski, B. J., Tsurutani, B. T., Lakhina, G. S. & Pickett, J. S. Two sources of dayside intense, quasi-coherent plasmaspheric hiss: A new mechanism for the slot region?. *J. Geophys. Res. Space Phys.* **122**, 1643–1657 (2017).
- Li, J. et al. Comparison of formulas for resonant interactions between energetic electrons and oblique whistler-mode waves. *Physics of Plasmas.* **22**, 052902 (2015).
- Li, J., et al. Interactions between magnetosonic waves and radiation belt electrons: Comparisons of quasi-linear calculations with test particle simulations. *Geophys. Res. Lett.*, **41**, 4828– 4834 (2014).
- Li, W. et al. Statistical properties of plasmaspheric hiss derived from Van Allen Probes data and their effects on radiation belt electron dynamics. *J. Geophys. Res. Space Phys.* **120**, 3393-3405 (2015).
- Ma, Q., Mourenas, D., Li, W., Artemyev, A. & Thorne, R. M. VLF transmitters from ground-based transmitters observed by the Van Allen Probes: Statistical model and effects on plasmaspheric electrons. *Geophys. Res. Lett.* **44**, 6483-6491 (2017).
- Tao, X., J. Bortnik, J. M. Albert, and R. M. Thorne, Comparison of bounce-averaged quasi-linear diffusion coefficients for parallel propagating

whistler mode waves with test particle simulations, *J. Geophys. Res.*, 117, A10205 (2012).

- Tsurutani, B. T., Falkowski, B. J., Pickett, J. S., Santolik, O. & Lakhina, G. S. Plasmaspheric hiss properties: Observations from Polar. *J. Geophys. Res. Space Phys.* 120, 414–431(2015).
- Tsurutani, B. T. et al. Plasmaspheric hiss: Coherent and intense. *J. Geophys. Res. Space Phys.* 123, 10009S10029 (2018).
- Tsurutani, B. T. et al. Low frequency ($f < 200$ Hz) polar plasmaspheric hiss: coherent and intense. *J. Geophys. Res. Space Phys.* 124, 10063– 10084 (2019).

The electron energy being examined should be mentioned in the title of the paper. Most people think of the “electron gap” as being due to 100s of keV electrons, not these lower energy electrons. This should also be mentioned more prominently in the body of the paper. The higher energy electron slot is due to the combination of coherent (?) magnetosonic waves and coherent hiss? Some crude approximations are given in the 2019 paper above.

Reply:

We agree with the reviewer and that’s why we didn’t call it “radiation belt”, but “energetic electron belt” in the manuscript, to distinguish the energetic electron populations in our study from the relativistic (MeV) electron populations. We have emphasized the specific electron energy range (tens of keV) throughout the main text. Please see Lines 25, 68, 70, 77, 87, 122, 134, 164, 184, 187, and 191 – 192. However, since the title has space limit and “tens of kiloelectron volts electron belt” sounds a bit wordy, we prefer not to include this detailed energy limit in the title.

Plasmaspheric hiss is known to dominantly drive electron pitch angle scattering for the higher energies ($> \sim 100$ keV) in the slot region between the inner and outer radiation belt, different from the VLF transmitter waves that drive the electron scattering at tens of keV energies leading to the bifurcation of the energetic electron belt. A number of studies have demonstrated that magnetosonic waves are capable of accelerating radiation belt electrons and producing electron butterfly pitch-angle distributions via the Landau resonance (e.g., Horne et al., 2007; Xiao et al., 2015; Li et al., 2016). Therefore, although magnetosonic waves can contribute to pitch angle scattering, plasmaspheric hiss still plays the dominant role in forming the slot region between the inner and outer belts at $> \sim 100$ keV (e.g., Lyons and Thorne, 1973; Falkowski et al., 2017; Ma et al., 2016). We have added the discussion that hiss dominantly drives the slot region at higher energies ($> \sim 100$ keV) between the inner and outer radiation belts and added the related references given above. Please see Lines 138 – 139.

Reference:

- Falkowski, B. J., Tsurutani, B. T., Lakhina, G. S. & Pickett, J. S. Two sources of dayside intense, quasi-coherent plasmaspheric hiss: A new mechanism for the slot region?. *J. Geophys. Res. Space Phys.* **122**,1643–1657 (2017).

- Horne, R. B. et al. Electron acceleration in the Van Allen radiation belts by fast magnetosonic waves. *Geophys. Res. Lett.* **34**, S09S03 (2007).
- Xiao, F. et al., Wave-driven butterfly distribution of Van Allen belt relativistic electrons. *Nature Comm.* **6**, 8590 (2015).
- Li, J. et al. Formation of energetic electron butterfly distributions by magnetosonic waves via Landau resonance. *Geophys. Res. Lett.* **43**, 3009–3016 (2016).
- Lyons, L. R., and Thorne, R. M., Equilibrium structure of radiation belt electrons, *J. Geophys. Res.*, 78(13), 2142– 2149 (1973).
- Ma, Q., et al. Characteristic energy range of electron scattering due to plasmaspheric hiss, *J. Geophys. Res. Space Physics*, **121**, 11,737– 11,749 (2016).

Minor Comments

Lines 39-40. What does “when a magnetic field line crosses the geomagnetic equator” mean and how does this geometry affect transmitter waves allowing them to penetrate though the ionosphere? It would be good for the reader if you would elaborate a bit more and give references.

Reply:

We thank the reviewer for this helpful comment. We have rephrased the sentence as follows. “where L is the geocentric distance in Earth radii of the location where the corresponding magnetic field line crosses the geomagnetic equator.” While propagating mostly within the Earth-ionosphere waveguide, which is bounded by the terrestrial surface and the lower ionosphere at altitudes about ninety kilometers, VLF transmitter signals can penetrate through the imperfectly reflecting ionosphere, being guided by the gradients of the Earth’s magnetic field, to leak a portion of their power into the Earth’s magnetosphere primarily at $L < 3^{5-8}$. We have added this point and the related references. Please see Lines 38 – 42.

Reference:

5. Helliwell, R. A. *Whistlers and Related Ionospheric Phenomena* (Stanford Univ. Press, Stanford, Calif, 1965).
6. Cohen, M. B., & Inan, U. S. Terrestrial VLF transmitter injection into the magnetosphere. *J. Geophys.Res.* **117**, A08310 (2012).
7. Starks, M. J. et al. Illumination of the plasmasphere by terrestrial very low frequency transmitters: Model validation, *J. Geophys. Res.* **113**, A09320 (2008).
8. Clilverd, M. A. et al. Ground-based transmitter signals observed from space: Ducted or nonducted?. *J. Geophys. Res.* **113**, A04211 (2008).

Lines 54-56. The energy of ~30 to 200 keV is beyond the limit of your study, so how is this relevant?

Reply:

The bifurcation of energetic electron belts shown in Claudepierre et al. (2020) is obvious at energies over 32 – 132 keV, while the electrons at higher energies do not show the bifurcation features. We have rephrased this description as follows.

“a recent study showed the bifurcation of energetic electron belt at energies of ~30–130 keV over L -shells of 2–3.” Please see Line 57 – 58.

Reference:

Claudepierre, S. G. et al. Empirically estimated electron lifetimes in the Earth's radiation belts: Comparison with theory. *Geophys. Res. Lett.* **47**. (2020).

Lines 76-77. Moderate substorms. Chorus is generated during substorms. It has been hypothesized that chorus propagates into the plasmasphere. This seems like a reasonable competing mechanism to VLF transmitters?

Reply:

We agree with the reviewer that a portion of chorus waves can propagate from outside the plasmapause into the high-density plasmasphere to form plasmaspheric hiss (e.g., Bortnik et al., 2008, 2009), the effect of which has already been considered in the present study, as shown in Figure 3. During the quiet event analyzed in the present study, chorus waves are typically observed at larger L -shells (> 3) (e.g., Meredith et al., 2012), thus it is unlikely that chorus can directly scatter energetic electrons to contribute to the formation of the bifurcated electron belt at $L < 2.5$. On the contrary, VLF transmitter signals can be continuously observed at low L -shells (< 3) for a long time period, thus can efficiently scatter electrons for a relatively long time. Therefore, electron scattering by VLF transmitter waves is still the most likely mechanism for the formation of the bifurcated energetic electron belts at energies of tens of keV.

Reference:

- Bortnik, J., R. M. Thorne, and N. P. Meredith, The unexpected origin of plasmaspheric hiss from discrete chorus emissions, *Nature*, 452(7183), 62–66 (2008).
- Bortnik, J., W. Li, R. M. Thorne, V. Angelopoulos, C. Cully, J. Bonnell, O. Le Contel, and A. Roux, An observation linking the origin of plasmaspheric hiss to discrete chorus emissions, *Science*, Vol. 324, Iss. 5928, p. 775 (2009).
- Meredith, N. P., R. B. Horne, A. Sicard-Piet, D. Boscher, K. H. Yearby, W. Li, and R. M. Thorne, Global model of lower band and upper band chorus from multiple satellite observations, *J. Geophys. Res.*, 117, A10225 (2012).

Lines 102-109. I don't understand the discussion here. If these natural waves (lightning, hiss and magnetosonic waves) “significantly exceed the VLF transmitter intensity”, then why do the simulations indicate that it is transmitter signals that are causing the slot? Perhaps what you mean is that at VLF frequencies the transmitter intensities have the highest intensities? Or have the highest intensities in the VLF frequency range? Something seems to be missing here.

Reply:

We thank the reviewer for pointing this out. Indeed, the VLF transmitter wave power intensity at ~24 kHz is higher than the intensity of hiss, lightning generated waves and magnetosonic waves at the same frequency, although the integrated wave amplitude

of VLF transmitter wave is weak. Since the wave amplitude of lightning generated whistlers is similar to that of VLF transmitter waves, we replaced ‘significantly exceed the VLF transmitter intensity’ with ‘exceed or become comparable to the VLF wave amplitude’. Please see Lines 118.

One important reason that VLF transmitter waves dominantly drive the bifurcated energetic electron belt at tens of keV is that the resonant electron energies due to VLF transmitter waves match the energies where the most significant electron flux decay is observed (see the white lines in Fig. 1f-h and Fig. R1 below). We have added a sentence to clarify this point in Lines 119 - 122. “However, the energy of electrons exhibiting the most evident bifurcation feature is close to the first-order cyclotron resonance energy corresponding to the waves at high frequencies (> 10 kHz), suggesting the potentially dominant role of VLF transmitter waves in bifurcating the energetic electron belt at tens of keV.”

Figure R1. The first-order cyclotron resonant energies of electrons interacting with plasmaspheric hiss with frequency of 252 Hz (red) and 4000 Hz (blue), and with VLF transmitter waves at 24 kHz (black) for 0° pitch angle at the magnetic equator.

Lines 119-121. Here you mention hiss is responsible for the high energy electron slot. Please discuss the possibility of either hiss Landau interactions with the 10s of keV electrons or the high frequency ends of hiss for cyclotron resonance. This is a bit confusing.

Reply:

We thank the reviewer for this constructive comment. Plasmaspheric hiss can contribute to scatter tens of keV electrons at pitch angles close to 90° by Landau resonance, but the scattering becomes ineffective at the pitch angles below $\sim 60^\circ$. The Landau resonance and cyclotron resonances (up to 10) due to hiss are included in our simulations. This point is discussed in Lines 143 – 145 and 264 – 265.

Our diffusion rate calculation includes hiss power at frequencies up to 4 kHz, but the peak wave power is observed below a few hundred Hz, and the hiss wave power at

higher frequencies above ~1 kHz is extremely weak (Li et al., 2015). The pitch-angle diffusion coefficients for plasmaspheric hiss start to become strong above ~100 keV near the bounce loss cone at $L < 2.3$ (see Fig. 3g & 3h), thus contribute little to electron scattering loss at energies < 100 keV. At $L > 2.3$ the pitch-angle diffusion coefficients due to hiss near the bounce loss cone tend to become strong below 100 keV. This trend is also shown in Fig. R1 above. Nevertheless, their effect on tens of keV electron bifurcation is less efficient than that of VLF transmitter waves, since the wave power of plasmaspheric hiss peaks at several hundred Hz. We have added this point to the main text. Please see Lines 143 – 145.

Reference:

Li, W. et al. Statistical properties of plasmaspheric hiss derived from Van Allen Probes data and their effects on radiation belt electron dynamics. *J. Geophys. Res. Space Phys.* **120**, 3393-3405 (2015).

Lines 124-126. Previous authors have discussed the combination of magnetosonic waves with plasmaspheric hiss for the pitch angle diffusion of energetic electrons, not magnetosonic waves by themselves. So this statement is a bit unfair.

Reply:

We thank the reviewer for pointing this out. Previous parametric study of Hua et al. (2019) has investigated the combined electron scattering effect by simultaneously occurring plasmaspheric hiss and magnetosonic waves with groups of different relative wave amplitude, which suggests that the combined scattering effects are dominated by pitch angle scattering due to hiss when hiss wave amplitude is comparable to or stronger than that of magnetosonic waves. In the present study, the wave amplitude of hiss is almost as twice as that of magnetosonic wave (shown in Fig. 2d in the main text). Therefore, the diffusion coefficients due to magnetosonic waves are much weaker than those of plasmaspheric hiss (shown in Fig. 3). Furthermore, Fig. 3s-u shows the combined effects of hiss, magnetosonic wave, and lightning generated whistlers, indicating that there is no clear formation of the bifurcating feature without including VLF transmitter waves. We have rephrased the sentence as follows on Lines 145 – 148. “The electron scattering rates due to magnetosonic waves are confined to high equatorial pitch-angles and are negligibly small near the bounce loss cone (Fig. 3j-l). Overall, the effects of magnetosonic waves on electrons are weakest compared to other three types of plasma waves.”

Reference:

Hua, M. et al. Evolution of radiation belt electron pitch angle distribution due to combined scattering by plasmaspheric hiss and magnetosonic waves. *Geophys. Res. Lett.* **46**, 3033–3042 (2019).

Line 132. Radial diffusion. Here are a couple of papers on this topic: GRL., 26, 3273, 1999; SW, 2, S10S02, doi:10.1029/2004SW000070., 2004. However this mechanism is

usually quoted for relativistic electrons, not the low energies that you are discussing. Could low energy electrons radially diffuse much by this process?

Reply:

We thank the reviewer for this valuable comment and introducing interesting papers, which are now cited on Line 158. The study of O'Brien et al. (2016) has estimated the quiet time radial diffusion coefficients for electrons in the inner radiation belt ($L < 3$) with energies from ~50 to 750 keV based on the Van Allen Probes observations, which agree well and supports the model of Brautigam and Albert (2000). For example, the radial diffusion coefficients based on measurements from the Van Allen Probes during quiet time ($Kp < 4$) is $\sim 10^{-3} \text{ day}^{-1}$ for $\mu = 1.9$ and 8.0 MeV/G at $L = 2.0$, while the radial diffusion coefficients from the model of Brautigam and Albert (2000) during quiet time ($Kp = 1$) is $\sim 2.5 \times 10^{-3} \text{ day}^{-1}$ for $\mu = 1.9$ MeV/G, and $\sim 10^{-3} \text{ day}^{-1}$ for $\mu = 8.0$ MeV/G at the same L shell region (see Figure 4 in O'Brien et al., 2016). Therefore, we think the model of Brautigam and Albert (2000) is a reasonable estimation of radial diffusion coefficients for low energy electrons in the inner belt during quiet times.

We have added this point in the Supplementary Information in Lines 158 – 160.

Reference:

- O'Brien, T. P. et al. Inner zone and slot electron radial diffusion revisited. *Geophys. Res. Lett.* **43**, 7301–7310 (2016).
- Brautigam, D. H. & Albert, J. M. Radial diffusion analysis of outer radiation belt electrons during the October 9, 1990, magnetic storm. *J. Geophys. Res.* **105**, 291–309 (2000).
- Elkington, S. R., Hudson, M. K., & Chan, A. A. (1999). Acceleration of relativistic electrons via drift-resonant interaction with toroidal-mode Pc-5 ULF oscillations. *Geophysical Research Letters*, 26(21), 3273– 3276. <https://doi.org/10.1029/1999GL003659>
- Miyoshi, Y. S., Jordanova, V. K., Morioka, A., and Evans, D. S. (2004), Solar cycle variations of the electron radiation belts: Observations and radial diffusion simulation, *Space Weather*, 2, S10S02, doi:10.1029/2004SW000070.

REVIEWER COMMENTS

Reviewer #1 (Remarks to the Author):

The authors have effectively resolved all of my previous comments, and have done a good job at addressing the numerous comments of the other reviewers. This manuscript was unusually good when first submitted, and has been strengthened by additional analysis, explanation and references. I believe it is suitable for publication in Nature Communications.

Reviewer #2 (Remarks to the Author):

I would like to thank the authors for their clear and concise responses both to my comments and the comments of the other reviewers and I can now recommend the paper for publication in Nature Communications.

Reviewer #3 (Remarks to the Author):

Second Review of "Human Made Very-Low Frequency Transmitters Bifurcate Energetic Electron Belt in Near-Earth Space" by Man, Li, Ni, Ma, Green, Shen, Claudepierre, Bortnik, Gu, Fu, Xiang, and Reeves

The authors have answered my and the other referees' queries and comments thoroughly. I am sure that the readers of the article will appreciate the improvement in the article. The responses were excellent!

I am satisfied with the paper and in my opinion it is acceptable for publication in Nature Communicates.

One final comment. The authors should note that recent hiss measurements indicate an order of magnitude higher amplitudes than what you have modeled, up to 0.3 nT. This is two orders of magnitude in intensity. However some of these waves are elliptically polarized and propagating off-axis. Both of these wave properties will reduce the scattering of the resonant electrons.

We thank all reviewers for careful reading of the manuscript and valuable and constructive comments to improve the quality of the paper. We have made point-by-point responses to the detailed comments by all reviewers and incorporated the suggested changes in the revised manuscript accordingly, as highlighted in the blue color.

Responses to Reviewer #1:

The authors have effectively resolved all of my previous comments, and have done a good job at addressing the numerous comments of the other reviewers. This manuscript was unusually good when first submitted, and has been strengthened by additional analysis, explanation and references. I believe it is suitable for publication in Nature Communications.

We thank the reviewer for the careful reading and the positive comment on our manuscript.

Responses to Reviewer #2:

I would like to thank the authors for their clear and concise responses both to my comments and the comments of the other reviewers and I can now recommend the paper for publication in Nature Communications.

We thank the reviewer for the careful reading and the positive comment on our manuscript.

Responses to Reviewer #3:

Second Review of "Human Made Very-Low Frequency Transmitters Bifurcate Energetic Electron Belt in Near-Earth Space" by Man, Li, Ni, Ma, Green, Shen, Claudepierre, Bortnik, Gu, Fu, Xiang, and Reeves

The authors have answered my and the other referees' queries and comments thoroughly. I am sure that the readers of the article will appreciate the improvement in the article. The responses were excellent!

I am satisfied with the paper and in my opinion it is acceptable for publication in Nature Communicates.

One final comment. The authors should note that recent hiss measurements indicate an order of magnitude higher amplitudes than what you have modeled, up to 0.3 nT. This is two orders of magnitude in intensity. However some of these waves are elliptically polarized and propagating off-axis. Both of these wave properties will reduce the scattering of the resonant electrons.

We thank the reviewer for the positive and this valuable comment, which helped us improve the quality of our paper. We agree with the reviewer that dayside strong hiss event with wave amplitudes up to a few hundred pT can be observed with elliptical polarization and oblique propagation, which could potentially drive sporadic electron precipitation (Falkowski et al., 2016). Nevertheless, the gradual decay profiles of electron fluxes in this event were observed during a relatively long and quiet time period (~15 days), and thus the statistical hiss wave model (Li et al., 2015) likely

provides a reasonable estimate on the effects of hiss waves on energetic electron scattering.

We further evaluated the wave amplitude of plasmaspheric hiss during this event. Fig. R1 shows the radial profile of daily-averaged magnetic wave amplitudes of plasmaspheric hiss measured by both Van Allen Probes from 21 February to 06 March 2016. Plasmaspheric hiss was selected over the frequency range of 20 – 4000 Hz in the plasmasphere with ellipticity larger than 0.7. Overall, the amplitudes of the observed hiss waves during this event were mostly below 30 pT at the low L -shells (< 2.5) corresponding to the most significant electron flux decay, which agrees well with the statistical results from Li et al. (2015). An example of the plasmaspheric hiss event observed by Van Allen Probe A on 23 February 2016 is shown in Fig. R2 below. As indicated by Fig. R1, hiss wave intensity was relatively strong on 23 February compared to other days, but the wave amplitude at L -shells over 1.5 – 3.0 (shown as the red lines in Fig. R2f) was overall below 40 pT. Therefore, we think it is reasonable to adopt the statistical wave amplitude of hiss in our simulations.

We agree with the reviewer that the off-axis propagating component of hiss waves can reduce the scattering of electrons. Recent statistical study of Hartley et al. (2018) demonstrates that plasmaspheric hiss over $L = 1.5 - 3.0$ mostly undergoes quasi-field-aligned propagation with wave normal angles (WNA) mostly less than $\sim 30^\circ$, while the oblique secondary wave population with WNA reaching $\sim 80^\circ$ has a weaker magnetic field intensity. Meredith et al. (2006) suggests that plasmaspheric hiss with small or intermediate WNA is dominantly responsible for electron loss while the hiss with large WNA does not contribute significantly to the loss rates. Furthermore, recent parametric studies (Ripoll et al., 2014; Gao et al., 2015) indicate that the loss rate due to hiss generally becomes less efficient with increasing WNA.

We have added these points and discussions in the paper. Please see Lines 83-86, 120-125, and 147-149 in the Supplementary Information.

Reference:

- Falkowski, B. J., Tsurutani, B. T., Lakhina, G. S. & Pickett, J. S. Two sources of dayside intense, quasi-coherent plasmaspheric hiss: A new mechanism for the slot region?. *J. Geophys. Res. Space Phys.* 122,1643–1657 (2017).
- Li, W. et al. Statistical properties of plasmaspheric hiss derived from Van Allen Probes data and their effects on radiation belt electron dynamics. *J. Geophys. Res. Space Phys.* 120, 3393-3405 (2015).
- Hartley, D. P., Kletzing, C. A., Santolík, O., Chen, L., & Horne, R. B. Statistical properties of plasmaspheric hiss from Van Allen Probes observations. *Journal of Geophysical Research: Space Physics*, 123, 2605– 2619 (2018).
- Meredith, N. P., Horne, R. B., Glauert, S. A., Thorne, R. M., Summers, D., Albert, J. M., & Anderson, R. R. Energetic outer zone electron loss timescales during low geomagnetic activity, *J. Geophys. Res.*, 111, A05212 (2006).

- Ripoll, J.□F., Albert, J. M., and Cunningham, G. S. Electron lifetimes from narrowband wave□particle interactions within the plasmasphere, J. Geophys. Res. Space Physics, 119, 8858– 8880 (2014).
- Gao, Y., Xiao, F., Yan, Q., Yang, C., Liu, S., He, Y., and Zhou, Q. Influence of wave normal angles on hiss□electron interaction in Earth's slot region, J. Geophys. Res. Space Physics, 120, 9385– 9400 (2015).

Figure R1. The radial profile of daily-averaged magnetic wave amplitude of plasmaspheric hiss over 20 – 4000 Hz based on the WFR measurements from both Van Allen Probes from 21 February to 06 March 2016.

Figure R2. *In situ* observations of plasmaspheric hiss by Van Allen Probe A on 23 February 2016. **a**, Frequency-time spectrogram of electric spectral density measured by HFR. **b**, Frequency-time spectrogram of wave spectral density in electric field and **c**, magnetic field observed by WFR, and the corresponding **d**, wave normal angles, **e**, ellipticity and **f**, integrated wave amplitude of plasmaspheric hiss with the red lines indicating the amplitudes at L -shells over 1.5 – 3.0. In Figs. R2a-e, the solid, dash-dotted, and dashed magenta lines indicate f_{ce} , $0.5 f_{ce}$, and $0.1 f_{ce}$, where f_{ce} is the electron cyclotron frequency; the solid, dotted, and dashed black lines represent f_{LHR} , $0.5 f_{LHR}$, and f_{cp} , where f_{LHR} is the lower hybrid resonance frequency and f_{cp} is the proton cyclotron frequency.

REVIEWERS' COMMENTS:

Reviewer #3 (Remarks to the Author):

Third Review of "Human-Made Very Low-Frequency Transmitters Bifurcate Energetic Electron Belt in Near-Earth Space" by Man, Li, Ni, Ma, Green, Claudepierre, Bortnik, Giu, Fu, Xiang and Reeves

I had accepted the paper in my last review, so I was surprise to see this detailed response to my "aside" in this recent "reply".

However now that I am asked to reply once more, I want to make sure that it is understood that I was referring to the maximum hiss wave intensities during substorms. There are two more recent papers on plasmaspheric hiss (JGRSP, 123, <https://doi.org/10.1029/2018JA025975>, 2018; JGRSP, 124, <https://doi.org/10.1029/2019JA027102>, 2019) which are claiming that these coherent and intense waves are causing the high energy electron slot. This is different than the lower energy electrons that you are discussing in your paper. So some clarification is perhaps needed. Maybe putting an energy range in your title? Also the plasmaspheric wave intensities that you have quoted, are those intensities for the hiss waves cyclotron resonant with your electrons? This should be clarified for the readership.

We thank the reviewer for the valuable comments to improve the quality of the paper. We have made point-by-point responses to the detailed comments by the reviewer and incorporated the suggested changes into the revised manuscript accordingly, as highlighted by the “track-change” function.

Reviewer #3 (Remarks to the Author):

Third Review of “Human-Made Very Low-Frequency Transmitters Bifurcate Energetic Electron Belt in Near-Earth Space” by Man, Li, Ni, Ma, Green, Claudepierre, Bortnik, Giu, Fu, Xiang and Reeves

I had accepted the paper in my last review, so I was surprised to see this detailed response to my “aside” in this recent “reply”.

However now that I am asked to reply once more, I want to make sure that it is understood that I was referring to the maximum hiss wave intensities during substorms. There are two more recent papers on plasmaspheric hiss (JGRSP, 123, <https://doi.org/10.1029/2018JA025975>, 2018; JGRSP, 124, <https://doi.org/10.1029/2019JA027102>, 2019) which are claiming that these coherent and intense waves are causing the high energy electron slot. This is different than the lower energy electrons that you are discussing in your paper. So some clarification is perhaps needed. Maybe putting an energy range in your title? Also the plasmaspheric wave intensities that you have quoted, are those intensities for the hiss waves cyclotron resonant with your electrons? This should be clarified for the readership.

We thank the reviewer for these valuable comments. We agree with the reviewer that intense plasmaspheric hiss can play an important role in causing the high energy electron slot region, which has already been mentioned and discussed in the main text. Please see Lines 137 – 140. Besides, we have already cited these two references (Tsurutani et al., 2018 & 2019) in the main text and Supplementary Information. Please see Lines 137 –154 in the main text and Lines 80 – 83 in the Supplementary Information.

Furthermore, we have significantly emphasized the specific electron energy range (tens of keV) of the bifurcated energetic electron belts throughout the main text and Supplementary Information. Since there is a word limit (<15 words) for the title, we prefer not to include “tens of kiloelectron volts electron belt” in the title.

Moreover, the strong plasmaspheric hiss with intensities up to several hundred pT that we have cited (Falkowski et al., 2016) includes the hiss wave power over the entire hiss frequency band. These hiss waves can interact with electrons through cyclotron resonances, which are included in our calculation of diffusion coefficients. These points have already been included in the Supplementary Information on Lines 145 – 147, and Supplementary Table 1.

References:

- Falkowski, B. J., Tsurutani, B. T., Lakhina, G. S. & Pickett, J. S. Two sources of dayside intense, quasi-coherent plasmaspheric hiss: A new mechanism for the slot region?. *J. Geophys. Res. Space Phys.* 122,1643–1657 (2017).
- Tsurutani, B. T. et al. Plasmaspheric hiss: Coherent and intense. *J. Geophys. Res. Space Phys.* 123, 10009S10029 (2018).
- Tsurutani, B. T. et al. Low frequency ($f < 200$ Hz) polar plasmaspheric hiss: coherent and intense. *J. Geophys. Res. Space Phys.* 124, 10063– 10084 (2019).